# The Crowded Embedding Space: A Mean-Field Mechanism for Emergent Marginalization in Retrieval-Augmented Agents

**Shwan Ashrafi**[1]  **Dan Roth**[1]

## Abstract

Retrieval-augmented generative agents rely on retrieval for grounding, yet are typically evaluated on a query-by-query basis. This isolates interactions that are geometrically coupled in a shared embedding space. For example, we show that the high document density required to serve majority interests (e.g., generic "Crime" movies) can geometrically overcrowd the retrieval neighborhood of a semantically similar minority (e.g., "Film Noir"), effectively expelling minority content from top-$k$ results. We introduce a formal framework to analyze how such *goal collisions* in dense retrieval induce fundamental performance limits and emergent fairness issues inherent to spatial crowding. In our static analysis, we demonstrate that for a fixed embedding space, a phase transition occurs where minority user goals suffer a catastrophic collapse in performance as the density of majority goals increases. We then extend this to a dynamic model and derive a non-linear Fokker-Planck equation that governs the evolution of document embeddings as the agent updates them to maximize retrieval accuracy. Our analysis reveals that this local relevance objective triggers an emergent global mechanism that systematically marginalizes minority interests. We prove that such objectives drive the system to self-organize into a state that exclusively serves majority interests. These results provide a theoretical foundation for understanding a critical grounding failure mode in retrieval-augmented agents.

## 1. Introduction

Large language models (LLMs) are increasingly deployed as agents that interact with users and external systems, often

[1]Oracle AI. Correspondence to: Shwan Ashrafi <shwan.ashrafi@oracle.com>.

*Proceedings of the 43rd International Conference on Machine Learning*, Seoul, South Korea. PMLR 306, 2026. Copyright 2026 by the author(s).

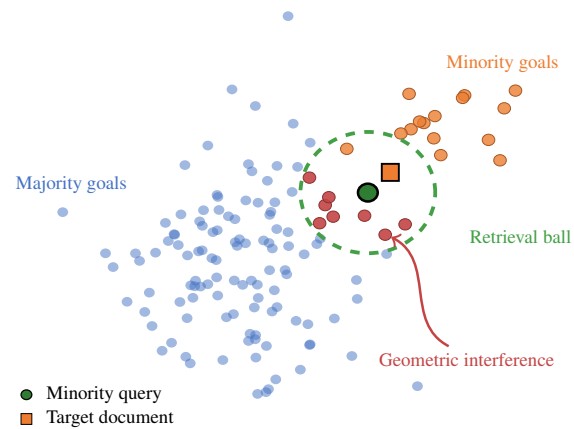

*Figure 1.* **The Geometry of Goal Collision.** Illustration of the interference mechanism for a minority query (green circle) seeking its target (orange square). As the density of the surrounding majority document population (blue points) increases, interfering documents (red points) statistically saturate the local neighborhood (green dashed circle). This geometric crowding effectively pushes the target out of the top-$k$ shortlist. We show that this local spatial saturation leads to a highly non-linear collapse in retrieval probability as majority density scales (Theorem 2.2).

relying on retrieval-augmented generation (RAG) to ground their responses in large, continuously evolving knowledge bases (Lewis et al., 2020; Guu et al., 2020; Ram et al., 2023; Kishore et al., 2023). Considerable progress has been made in building such systems which has resulted in improving retrieval accuracy, scaling vector search, and integrating reranking and feedback mechanisms. Yet, despite these advances, a fundamental understanding of how retrieval-augmented agents behave in realistic, multi-user environments remains critically lacking.

While the agent's final output may benefit from parametric memory or redundancy, the core premise of RAG is that reliable grounding requires the retrieval of specific, external evidence. The dominant evaluation paradigm for RAG systems assesses performance on isolated, independently sampled queries (Chen et al., 2024). This paradigm implicitly assumes that each query can be evaluated in isolation, independent of other users or queries. In practice, however, deployed agents are shared resources. A single retrieval system simultaneously serves a heterogeneous population, necessitating a shared index that represents the union of

all diverse user interests. Retrieval outcomes are not only subject to algorithmic fairness constraints (Kim & Diaz, 2025), but are also driven by population dynamics, where users collectively shape which documents are retrieved and reinforced over time (Morik et al., 2020). As a result, the grounding capacity of any individual user query is no longer an isolated event, but is coupled, often implicitly, to the behavior of the broader user population.

At the core of every RAG system lies a high-dimensional embedding space in which both queries and documents are represented as vectors. Retrieval is performed by nearest-neighbor search in this space. While such representations are highly expressive, they are also subject to fundamental geometric constraints. As the number of distinct user goals grows, the finite dimensionality and effective anisotropy of embedding spaces guarantee that neighborhoods corresponding to different goals will increasingly overlap (Valeriani et al., 2023). We refer to this phenomenon as *goal collision* (Figure 1). When a user with a niche or low-frequency goal issues a query, their nearest neighbors are likely to include documents relevant to more frequent or popular goals that happen to occupy the same region of the embedding space (Liu et al., 2024). For minority queries where redundancy is low and parametric knowledge is unreliable, this exclusion from the retrieval shortlist effectively creates a hard bottleneck for the agent. This contamination is not due to algorithmic error, but emerges endogenously from geometric crowding, where the high density of majority data saturates the retrieval space, imposing a global proximity scale that is too coarse to resolve the finer local separation of minority clusters.

Goal collision introduces an implicit population-level dependency. Specifically, a given user's retrieval performance depends on the density of *other* users; this occurs because distinct goals compete for finite retrieval capacity, where dense majority populations statistically exclude semantically adjacent minority targets from the shortlist by monopolizing the nearest-neighbor ranks. This phenomenon connects retrieval performance to broader concerns in fairness and exposure, but differs from classical fairness analyses in that it occurs even when the system faithfully learns the underlying population statistics (Singh & Joachims, 2018).

In this paper, we argue that this implicit dependency is not merely a source of stochastic noise or variance, but a mechanism for *emergent system-level behavior*. We develop a theoretical framework, grounded in mean-field theory (Mei et al., 2018), to analyze the population-level consequences of goal collision in retrieval-augmented agents. Our framework treats user goals and document embeddings as interacting populations within a shared geometric space, allowing us to move beyond per-query analysis and reason about macroscopic system behavior.

Our contributions are twofold. First, we present a static analysis of retrieval in a fixed embedding space. We formally relate the local density of user goals to the probability of successful retrieval and show that this relationship is highly non-linear. In particular, we prove the existence of a sharp *phase transition* in which, as the density of majority goals surpasses a critical threshold, the retrieval success rate for minority goals collapses catastrophically, rendering them effectively invisible to the agent. This collapse is not gradual, but abrupt, and is amplified in practical two-stage retrieval pipelines with finite shortlist budgets. Our analysis provides a theoretical explanation for empirically observed phenomena such as hubness and the systematic exclusion of low-density regions in embedding-based retrieval (Nielsen & Hansen, 2023).

Second, we extend our analysis to the dynamic setting in which retrieval-augmented agents undergo retraining or fine-tuning of their retrieval components from user feedback, inducing an effective evolution of the document embedding distribution over time (Vu et al., 2024). We model this learning process using a mean-field McKean–Vlasov equation that governs the macroscopic evolution of the document embedding density (Mignacco et al., 2020). This formulation reveals an emergent mechanism by which standard accuracy-maximizing objectives trigger a global reallocation of representational capacity within the shared embedding space. We prove that the system self-organizes to exclusively serve majority interests, concentrating embeddings around dominant goal regions while dynamically erasing minority interests.

These results provide a theoretical foundation for understanding a critical failure mode of retrieval-augmented agents in shared environments. Our findings show that emergent unfairness can result from geometric constraints and feedback-driven learning dynamics, under standard accuracy-maximization objectives. This mechanism is analogous in spirit to model collapse in generative systems trained on their own outputs (Shumailov et al., 2023; Dohmatob et al., 2024), but operates through retrieval geometry and population dynamics rather than data recursion. In this paper, we lay the groundwork for principled detection and mitigation strategies in the next generation of agentic AI systems.

## 2. The Static Model: Performance Limits from Goal Collision

We begin by analyzing a static setting in which both user goals and document embeddings are fixed. This abstraction isolates the geometric and population-level constraints inherent to embedding-based retrieval, allowing us to characterize fundamental performance limits that persist even in the absence of learning, feedback loops, or temporal effects.

Our goal in this section is not to model a specific retrieval system in full detail, but to capture the dominant mechanisms by which crowding in a shared embedding space degrades retrieval performance for low-density user goals.

## 2.1. Geometric Retrieval Model and Population Assumptions

Let the system be defined by a $d$-dimensional embedding space $\mathcal{S} \subseteq \mathbb{R}^d$. A population of users is characterized by a set of $N$ goals $\{g_1, \ldots, g_N\}$, which are mapped by an embedding function $\phi$ to points $\{\mathbf{g}_1, \ldots, \mathbf{g}_N\} \subset \mathcal{S}$. We assume that $N$ is large and that individual goals are drawn from an underlying population distribution. Accordingly, we model the empirical distribution of embedded goals using a continuous density $\rho_G(\mathbf{x})$ over $\mathcal{S}$, normalized such that $\int_{\mathcal{S}} \rho_G(\mathbf{x})\, d\mathbf{x} = 1$. The system's knowledge base consists of $M$ documents $\{d_1, \ldots, d_M\}$ with corresponding embeddings $\{\mathbf{d}_1, \ldots, \mathbf{d}_M\} \subset \mathcal{S}$. For simplicity, we assume a one-to-one correspondence where each goal $g_i$ has a single correct target document $d_i$, and their embeddings are initially close, i.e., $\phi(g_i) \approx \phi(d_i)$. This assumption allows us to cleanly define retrieval success and isolate interference effects. Importantly, our analysis does not assume semantic uniqueness or exclusivity of documents; instead, it considers a minimal setting in which each goal is associated with a well-defined target. Extensions to multi-document relevance or soft relevance do not qualitatively alter the geometric mechanisms we describe.

**Retrieval rule.** We consider a simple $k = 1$ nearest-neighbor search in the embedding space. A query corresponding to goal $g_i$, embedded at $\mathbf{g}_i$, is successful if and only if its target document $d_i$ (embedded at $\mathbf{d}_i$) is closer to $\mathbf{g}_i$ than any other document embedding $\mathbf{d}_j$, $j \neq i$. While real-world RAG systems often use approximate nearest-neighbor search and multi-stage pipelines, this minimal retrieval rule allows us to derive interpretable performance bounds. We later show that practical constraints such as shortlist truncation only amplify the effects identified here.

**Definition 2.1** (Success Probability)**.** The success probability for a query associated with a goal embedded at location $\mathbf{g}$ is defined as the probability that its target document at $\mathbf{d}$ is the nearest neighbor among all document embeddings $\mathcal{D} = \{\mathbf{d}_j\}_{j=1}^{M}$:

$$P_{\text{success}}(\mathbf{g}) = \mathbb{P}\left( \arg\min_{\mathbf{d}' \in \mathcal{D}} \|\mathbf{g} - \mathbf{d}'\|^2 = \mathbf{d} \right).$$

**Goal collision as geometric interference.** Retrieval failure occurs when at least one non-target document embedding $\mathbf{d}_j$ lies closer to the query $\mathbf{g}$ than the target embedding $\mathbf{d}$. Geometrically, this corresponds to the presence of an *interfering document* within the closed ball centered at $\mathbf{g}$

with radius $\|\mathbf{g} - \mathbf{d}\|$. We refer to such events as *goal collisions*, since they occur when documents from other user goals occupy the same local neighborhood of the embedding space. In the large-population regime, it is natural to reason about collisions statistically. Let $\lambda_D(\mathbf{x})$ denote the document intensity function. In many retrieval systems—particularly those where document embeddings are learned or refreshed based on user feedback or demand—the spatial distribution of documents closely tracks the distribution of user goals. As a first-order approximation, we therefore assume $\lambda_D(\mathbf{x}) \approx M\rho_G(\mathbf{x})$. In real-world static corpora (e.g., Wikipedia, the Web, or corporate knowledge bases), the supply of documents naturally mirrors historical human interest; mainstream topics organically accumulate larger volumes of written content than niche topics. Therefore, even a fixed snapshot of a corpus inherently possesses the skewed density distribution that triggers this geometric interference. This approximation is not exact, but it provides a tractable mean-field description that makes explicit how population density enters retrieval performance.

**Mean-field approximation.** Let $\epsilon = \|\mathbf{x} - \mathbf{d}\|$ denote the distance between a query location $\mathbf{x}$ and its target document. For sufficiently small $\epsilon$, the volume of the $d$-dimensional ball of radius $\epsilon$ is $V_d(\epsilon) = \frac{\pi^{d/2}}{\Gamma(d/2+1)}\epsilon^d$. The expected number of interfering documents within this ball is approximately $\lambda_D(\mathbf{x})V_d(\epsilon)$. Modeling document locations as a Poisson point process (PPP) with this local intensity, a standard approximation in stochastic geometry, yields a closed-form expression for the probability that no interferers are present. Under this approximation, the success probability at location $\mathbf{x}$ is given by

$$P_{\text{success}}(\mathbf{x}) \approx \exp(-\lambda_D(\mathbf{x})V_d(\epsilon)). \tag{1}$$

The uniform PPP assumption provides a conservative analytical lower bound; however, real-world document distributions are typically highly clustered. As discussed in Appendix B.1, this clustering produces pockets of super-critical density that accelerate performance collapse at substantially lower majority population thresholds. Furthermore, Equation (1) assumes a standard Euclidean volume, whereas practical neural embeddings (e.g., from Transformers) exhibit strong anisotropy and collapse into a narrow sub-manifold, commonly referred to as the "cone effect" (Appendix D). By substantially reducing the space's usable capacity, these real-world geometric constraints make spatial crowding and collisions inevitable even in high-dimensional spaces. Equation (1) forms the foundation of our static analysis. It explicitly links a *macroscopic* population property, the local density of user goals, to a *microscopic* retrieval outcome for an individual query. It can be seen that the dependence is exponential and even modest increases in local goal density can induce sharp drops in retrieval success for nearby

low-density goals. To illustrate the consequences of this exponential coupling, we consider a simple two-population setting in which a dense majority distribution coexists with a diffuse minority distribution. Even when the minority cluster is well separated and internally unchanged, growth in the majority population can induce a sharp collapse in minority retrieval success by setting a global collision scale, as illustrated by the Gaussian mixture example in Appendix B. Related geometric effects such as hubness and embedding anisotropy are discussed in Appendix D.

## 2.2. Theoretical Result: Static Performance Collapse

The abstract geometric crowding described by the exponential relationship is greatly amplified in practical, two-stage retrieval systems. We consider a system where a fast, first-stage retriever (e.g., Approximate Nearest Neighbor) returns a fixed-size shortlist of $L$ candidates, which are then processed by a second-stage reranker. In this budgeted setting, a retrieval fails if the correct document is not in the shortlist, leading to a much sharper performance collapse.

**Theorem 2.2** (Minority Collapse via Shortlist Exclusion). *Let a retrieval system serve a goal distribution $\rho_G(\mathbf{x}) = (1 - \alpha)\rho_{maj}(\mathbf{x}) + \alpha\rho_{min}(\mathbf{x})$ using a two-stage process with a first-stage shortlist of size $L$. For a minority query at location $\mathbf{x} \in \mathcal{S}_{min}$, let $V(\mathbf{x})$ be the volume of the retrieval ball containing its target document.*

*Define*

$$I_{\min} := \inf_{\mathbf{x} \in \mathcal{S}_{min}} \int_{V(\mathbf{x})} \rho_{maj}(z)\, dz,$$

*and let the critical majority population be*

$$N_c := \frac{L}{I_{\min}}.$$

*If $N_{maj} > N_c$, then for all $\mathbf{x} \in \mathcal{S}_{min}$, the probability that the correct document is included in the shortlist decays super-polynomially in $N_{maj}$, causing the expected success probability $\mathbb{E}_{\mathbf{x} \sim \rho_{min}}[P_{success}(\mathbf{x})]$ to collapse towards zero.*

**Corollary 2.3** (Multi-Document Relevance). *If a minority query has $T > 1$ valid targets and success requires capturing at least one in the top-$L$ shortlist, the threshold $N_c$ increases and the phase transition is delayed, but success probability still decays exponentially for $N_{maj} > N_c$.*

Theorem 2.2 identifies a sharp geometric phase transition driven by the majority document population. Below the critical threshold $N_c$, the retrieval mechanism operates in a coexistence regime where minority targets remain distinguishable within the shortlist. However, once the majority document population exceeds $N_c$, the local embedding neighborhood becomes saturated with interferers. This saturation causes the minority inclusion probability to vanish super-polynomially, resulting in a catastrophic collapse of retrieval

utility rather than gradual degradation. Theorem 2.2 establishes this behavior as a general bound. We complement this analysis with exact closed-form expressions for high-dimensional Gaussian mixtures in Appendix B.

# 3. The Dynamic Model: Emergent Marginalization

We now extend our analysis to the more realistic scenario where the RAG agent learns from user interactions over time. To map our formal derivations directly to practical RAG mechanics, consider that deployed systems increasingly use continuous index refreshing or fine-tuning based on user feedback. In this dynamic, standard gradient updates attempt to maximize local retrieval accuracy in a continuous, crowded space. We model this adaptive retrieval as a mean-field gradient flow, where accuracy-driven updates (the "drift") compete against stochastic learning noise (the "diffusion"). We will show that, in a shared multi-user environment, this learning process induces a mechanism of emergent unfairness. In particular, the system is driven toward a state of progressive representational collapse analogous to model autophagy in generative systems (Alemohammad et al., 2023).

## 3.1. Mean-Field Embedding Dynamics

We assume that the population of user goals is stationary over the time horizon of interest and is described by a fixed density $\rho_G(\mathbf{x})$ on the embedding space $\mathcal{S} \subseteq \mathbb{R}^d$. However, document embeddings evolve as the agent learns from interactions. Their distribution is described by a time-varying density $\rho_D(\mathbf{x}, t)$. This continuum approximation is appropriate when the number of documents is large and when updates are small but frequent.

**Learning as a gradient flow.** We model the learning dynamics (evolution of $\rho_D$) as the Wasserstein gradient flow of a free energy functional $\mathcal{F}[\rho_D]$. This formulation is standard in mean-field analyses of large interacting systems and captures the aggregate effect of many local updates driven by user feedback. The main modeling choice is the decomposition of $\mathcal{F}$ into two competing components, (i) an *interaction potential* that pulls document embeddings toward regions frequently queried by users (maximizing relevance), and (ii) an *entropic regularization* that reflects stochasticity in learning, exploration, and noise inherent in stochastic gradient descent updates.

We define the free energy as

$$\mathcal{F}[\rho_D] = \int_{\mathcal{S}} \left( \underbrace{K\,\rho_G(x)\,e^{-C\rho_D(\mathbf{x})}}_{\substack{\text{Interaction Potential} \\ \mathcal{V}[\rho_D]}} + \underbrace{D\,\rho_D(\mathbf{x})\ln\rho_D(\mathbf{x})}_{\substack{\text{Entropy} \\ \mathcal{E}[\rho_D]}} \right) d\mathbf{x}.$$

The first term $\mathcal{V}$, weighted by the learning rate $K$, encodes the agent's retrieval objective, and $C \equiv M \cdot V_d(\epsilon)$ represents the effective system capacity. Regions of the embedding space with high goal density $\rho_G(\mathbf{x})$ correspond to frequently queried intents. Because the term decays with $\rho_D$, minimizing this potential drives the system to concentrate document density in these regions to maximize retrieval success. The second term $\mathcal{E}$ is an entropic regularizer weighted by $D$. It penalizes highly concentrated document distributions and promotes exploration or noise in the learning process. The relative magnitude of $K$ and $D$ controls the balance between exploitation and exploration. We define the ratio $\beta = CK/D$ as the effective "learning pressure". In practical deployments, accuracy-driven updates typically dominate random SGD noise, driving the system into a high learning pressure limit ($\beta \to \infty$).

**Evolution equation.** The evolution of the document density $\rho_D(\mathbf{x}, t)$ is governed by the gradient flow of this functional under the Wasserstein metric. This yields the continuity equation:

$$\frac{\partial \rho_D}{\partial t} = \nabla \cdot \left( \rho_D \nabla \frac{\delta \mathcal{F}}{\delta \rho_D} \right). \tag{2}$$

By expanding the functional derivative $\frac{\delta \mathcal{F}}{\delta \rho_D}$, we recover the familiar drift–diffusion (Fokker–Planck) form. The variation of the entropy term yields the diffusion, while the variation of the potential yields the drift:

$$\frac{\partial \rho_D}{\partial t} = \nabla \cdot \left( \rho_D \nabla \left( \frac{\delta \mathcal{V}}{\delta \rho_D} + D(\ln \rho_D + 1) \right) \right)$$
$$= -\nabla \cdot (\mathbf{v}_{\text{drift}} \rho_D) + D\nabla^2 \rho_D,$$

where the effective drift velocity is $\mathbf{v}_{\text{drift}} = -\nabla \frac{\delta \mathcal{V}}{\delta \rho_D}$. This equation is non-linear because the drift depends explicitly on $\rho_D$ through the interaction potential.

### 3.2. Theoretical Result: Emergent Marginalization

We now characterize the long-term behavior of the learning dynamics by analyzing the steady-state solutions of the McKean–Vlasov equation (2), i.e., distributions $\rho_D^*$ satisfying $\partial \rho_D / \partial t = 0$. We show that the interaction between population skew and feedback-driven learning leads to an equilibrium where minority user goals are strictly excluded.

**Theorem 3.1** (Emergent Marginalization). *Let the goal distribution $\rho_G(\mathbf{x})$ have multiple disjoint local maxima (representing different topics or goal clusters). Assume that document embeddings evolve according to the gradient flow defined by Eq. (2), and that the learning rate $K$ be sufficiently large compared to the diffusion rate $D$. Then, as $t \to \infty$, the document density $\rho_D(\mathbf{x}, t)$ converges to a thresholded equilibrium:*

$$\rho_D^*(\mathbf{x}) = \max \left( 0, \frac{1}{C} \ln \left( \frac{\rho_G(\mathbf{x})}{\tau} \right) \right)$$

*where $\tau$ is a system-dependent critical threshold. Consequently, for any minority user group where the goal density $\rho_G(\mathbf{x}) \leq \tau$, the allocated document density is strictly zero, rendering these goals effectively invisible to the agent.*

Theorem 3.1 formalizes a mechanism of emergent marginalization. It proves that the agent, by following a locally optimal learning rule, reconfigures its knowledge base to serve only those user groups that exceed a critical viability threshold. While popular goals enjoy logarithmic representation, minority interests falling below this threshold are not merely underserved; they are actively and dynamically erased from the agent's operational capabilities. From a system-level perspective, this equilibrium represents a global minimum of the free energy functional $\mathcal{F}[\rho_D]$ where the agent optimizes for high-frequency goals while shedding capacity for low-frequency ones. This endogenous exclusion occurs without sampling bias or adversarial objectives, driven solely by the interaction between geometric crowding and the accuracy-maximizing update rule. We emphasize that while diffusion may sustain metastable coexistence in the short term, the asymptotic behavior in the high-learning-pressure regime is defined by this hard representational cutoff, leading to the delayed but irreversible collapse of minority embeddings observed in Section 4.4.

## 4. Experiments

We evaluate the theoretical predictions of Sections 2 and 3 through a combination of controlled synthetic simulations and empirical studies on real-world datasets. All experiments are designed to isolate population-induced geometric effects in fixed embedding spaces and to examine how standard learning dynamics amplify these effects over time. Our evaluation proceeds in two stages. First, we validate the existence and sharpness of the static phase transition predicted by Theorem 2.2 under controlled conditions. Second, we demonstrate that the same failure mode emerges consistently across text, vision, and narrative domains, and that learning dynamics governed by Theorem 3.1 transform this static fragility into irreversible marginalization.

### 4.1. Experimental Methodology

We use two pretrained models: (i) `all-MiniLM-L6-v2` for text encoding (384 dimensions), and (ii) `clip-ViT-B-32` for image encoding (512 dimensions). All embeddings are $L_2$-normalized to enable cosine similarity via dot product. For each experiment, we fix a small minority cluster $N_{\text{min}}$, and progressively

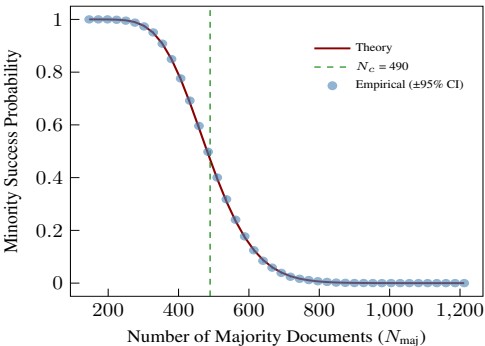

*Figure 2.* **Phase Transition.** Comparison of theoretical prediction (solid red line, Eq. (1)) against empirical simulation (grey dots) for minority retrieval success. As the number of interfering majority documents $N_{\text{maj}}$, exceeds the critical threshold $N_c$ (vertical dashed line), minority performance undergoes a catastrophic collapse. This confirms that geometric crowding acts as a hard constraint on retrieval capacity.

increase the size of an interfering majority population $N_{\text{maj}}$. We measure retrieval performance using Recall@$k$ for $k \in \{1, 2, 5, 10\}$, defined as the fraction of minority queries for which at least one relevant minority document appears in the top-$k$ retrieved results. To ensure statistical robustness, all experiments are repeated across multiple random seeds, and we report mean performance with 95% confidence intervals. The experimental protocol follows a leave-one-out consistency check in which, for each minority query embedding, we evaluate whether the retrieval system returns other minority documents before majority interferers. This design isolates geometric crowding effects by holding the minority cluster structure fixed while varying only the density of the surrounding majority population.

### 4.2. Synthetic Validation of the Phase Transition

We begin with a synthetic setting that directly instantiates the assumptions of Section 2. Majority documents are sampled from an isotropic Gaussian $\mathcal{N}(0, \sigma^2 I)$ in $\mathbb{R}^d$ with $d = 2$ and $\sigma = 1$. The minority query is placed at distance $1.5\sigma$ from the majority center, with its target at distance $\epsilon = 0.5$, and shortlist size $L = 20$. Figure 2 summarizes the results. As the size of the majority population $N_{\text{maj}}$ increases, the probability that at least $L$ interfering documents enter the retrieval ball of a minority query transitions sharply from near zero to near one. This induces a sudden collapse in retrieval success once the critical threshold $N_c$ predicted by Theorem 2.2 is exceeded. The empirical success probability closely follows the theoretical prediction, which shows that the collapse manifests as a sharp phase transition. We further observe that increasing the embedding dimensionality delays this transition but does not remove it, consistent with the closed-form analysis in Proposition B.1.

### 4.3. Universality of Phase Transition

We next evaluate whether the static phase transition persists in realistic embedding spaces produced by pretrained representation models. All experiments use fixed embeddings without retraining or fine-tuning, and retrieval is performed via exact nearest-neighbor search using cosine similarity. Across all datasets, we use a consistent experimental protocol in which a small, fixed minority cluster is exposed to increasing interference from a semantically or perceptually similar majority population. The internal structure of the minority cluster is held constant, and only the density of the majority population is varied. This setup isolates the population-level geometric effects predicted by our theory from confounding factors such as model quality or minority cluster coherence.

**Semantic Ambiguity (Text).** We use the Quora Question Pairs (QQP) dataset (Iyer et al., 2017) to model semantic interference. The dataset contains pairs of questions labeled as duplicates or non-duplicates. We sample $N_{\text{min}} = 100$ true duplicate pairs from the training split as the minority group, treating the first question as the query and the second as the target. The interfering population is constructed from up to 150,000 non-duplicate question pairs, which yields approximately 200,000–250,000 candidate interferers after deduplication. We exclude exact matches to minority targets to prevent accidental true positives. As shown in Figure 4d, Recall@1 for the minority remains near-perfect at low interference but collapses once $N_{\text{maj}} > 25000$. At $N_{\text{maj}} \approx 200,000$, Recall@1 drops below 0.6.

**Perceptual Crowding (Vision).** To test modality independence, we evaluate image retrieval using CLIP (ViT-B/32) on CIFAR-100 (Krizhevsky et al., 2009). We select "otter" as the minority class ($N_{\text{min}} = 50$) and use a leave-one-out protocol to check if the nearest neighbor is also an otter. The interfering population is constructed from 24 visually similar mammal classes, which results in approximately 12,000 images that are randomly shuffled. Figure 4a shows a sharp degradation in Recall@1 dropping from near-perfect to below 0.3 as $N_{\text{maj}}$ approaches 12,000.

**Narrative Style Preservation.** We evaluate stylistic retrieval using the Wikipedia Movie Plots dataset (Robischon, 2018), which contains movie plot descriptions annotated with genre labels. The minority consists of $N_{\text{min}} = 50$ "Film Noir" plots, and the majority includes generic "Crime/Mystery" plots. We measure whether each noir plot's nearest neighbor is also a noir plot. Interferers are drawn from three semantically overlapping but stylistically distinct genres ("action", "thriller", "crime"), excluding dual-labels. This yields $\approx 5000$ descriptions. As shown in Figure 4b, stylistic signals collapse at significantly lower interference

levels than semantic content. Recall@1 drops below 0.2 at $N_{\text{maj}} \approx 2000$. This sharp drop suggests that nuanced attributes like tone are particularly vulnerable to population-induced interference from high-volume mainstream content.

**Topic Retrieval on 20 Newsgroups.** We use the 20 Newsgroups dataset (Lang, 1995), which contains newsgroup posts across 20 topics, with "comp.sys.mac.hardware" designated as the minority topic. The standard train–test split is used, with documents from the training set forming the retrieval index and documents from the test set serving as queries. From these splits, $N_{\text{min}} = 100$ minority documents and $N_{\text{min}} = 100$ minority queries are selected. Ground-truth pairings are established by computing pairwise cosine similarities between all queries and minority documents and assigning each query to its nearest minority document. Approximately 18000 documents from the remaining 19 topics constitute the interfering population. Figure 4c shows the same qualitative phase transition, where Recall@1 drops from near-perfect performance to roughly 0.2 as $N_{\text{maj}}$ increases to 15000.

The phase transition behavior observed in these experiments aligns with the decay predicted by Theorem 2.2 and emerges despite the minority cluster structure remaining fixed. Notably, the performance collapse in these real-world datasets occurs much more rapidly—and at significantly lower majority population thresholds—than in the uniform synthetic setting (Figure 2). As detailed in Appendix B.1, this accelerated drop is precisely driven by the natural clustering of real-world documents. Clustering creates pockets of super-critical density, causing the local intensity to cross the interference threshold and trigger geometric exclusion much earlier than uniform point processes predict. Higher $k$ values provide only modest improvements, indicating that the geometric crowding effect is not easily mitigated by expanding the retrieval budget. The consistency of this pattern across four modalities (semantic text, visual, stylistic text, and topical text) provides compelling empirical evidence for the universality of the geometric crowding mechanism.

**Impact of Two-Stage Pipelines.** To determine if reranking mitigates geometric collapse, we evaluate a standard pipeline (SBERT retriever with MS-MARCO Cross-Encoder) on the 20 Newsgroups dataset, varying the shortlist size $L$ (Figure 3). We identify a dual failure mode for minority queries. First, restricted shortlist budgets create a hard retrieval bottleneck where even an Oracle reranker fails, empirically validating the exclusion bounds predicted by Theorem 2.2. Second, relaxing this constraint saturates ($L > 100$) the Cross-Encoder with a high density of "hard negatives" from the minority subspace. Consequently, actual ranking performance stagnates or degrades even as Oracle recall improves, demonstrating that geometric crowding

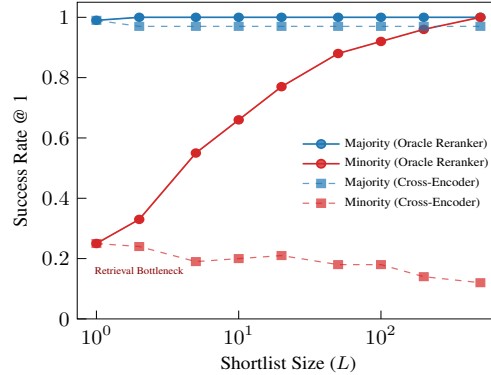

*Figure 3.* **Reranking Cannot Rescue Geometric Collapse.** Retrieval performance on 20 Newsgroups with increasing shortlist size $L$. An Oracle reranker (solid lines) benefits from larger shortlists, however, a realistic Cross-Encoder (dashed lines) fails to recover minority performance. The embedding space becomes so saturated with majority "hard negatives" that the downstream ranker cannot distinguish the true minority target, demonstrating that geometric crowding creates discriminative confusion robust to budget expansion.

induces downstream discriminative confusion that is robust to increased shortlist depth.

### 4.4. Dynamic Marginalization Under Learning

We investigate the hypothesis that standard learning dynamics drive the system from static fragility to irreversible marginalization. Using the Wikipedia Movie Plots dataset, we initialize a 384-dimensional embedding space with two well-separated clusters corresponding to "Film Noir" (minority, 50 plots) and "Crime/Mystery" (majority, 500 plots). Initial retrieval performance is near-perfect. We simulate user feedback with a 98% majority query rate. Embedding updates follow the drift–diffusion dynamics of Eq. (2) with learning rate $K = 0.05$ and additive Gaussian noise $\sigma = 0.002$, yielding a ratio $K/\sigma = 25$ that places the system in the learning-dominated regime of Theorem 3.1. At each step, documents whose cosine similarity to the sampled query exceeds an interaction threshold of $0.35$ are pulled toward the query centroid, with drift magnitude weighted by a softmax over similarities (temperature 10). All embeddings are re-normalized to the unit sphere after each update.

**Metastability and Collapse.** Figure 5 shows the temporal evolution of minority retrievability over 5000 simulation steps. We observe a distinct metastable regime ($t < 1500$) where minority Recall@1 for minority queries remains high ($> 0.8$) despite continuous drift. However, the inter-cluster $L_2$ centroid distance diminishes steadily throughout this phase. Upon crossing a critical interaction boundary where drift forces minority embeddings into the majority's high-density manifold, performance collapses catastrophically, dropping below 0.1 within 500 steps.

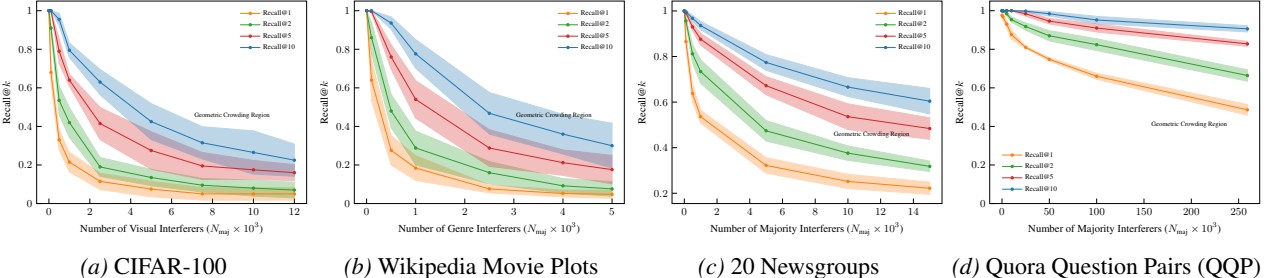

*(a)* CIFAR-100     *(b)* Wikipedia Movie Plots     *(c)* 20 Newsgroups     *(d)* Quora Question Pairs (QQP)

*Figure 4.* **Universality of Geometric Collapse.** We observe a consistent performance collapse across modalities as majority density increases. Shaded regions indicate 95% confidence intervals. The consistent degradation across $k \in \{1, 2, 5, 10\}$ demonstrates that increasing the retrieval budget provides negligible mitigation against density-induced exclusion. **(a) Visual Retrieval (CIFAR-100).** Perceptual crowding among semantically similar mammal classes causes recall to vanish. **(b) Stylistic Retrieval (Wikipedia Movie Plots).** Narrative style exhibits the most rapid collapse. This trend suggests that nuanced attributes are highly vulnerable to erasure by majority content. **(c) Topical Retrieval (20 Newsgroups).** High-density majority topics saturate the embedding space and displace the minority topic despite its thematic distinctness. **(d) Semantic Retrieval (QQP).** Lexical overlap drives collision between duplicate and non-duplicate questions.

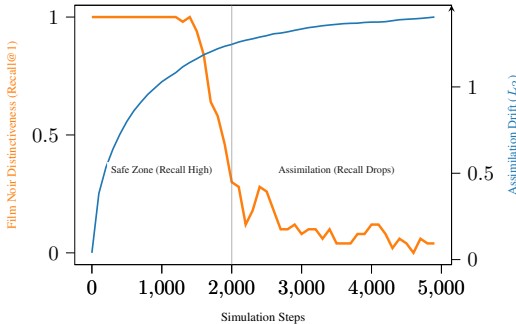

*Figure 5.* **Metastable Collapse of Fairness.** Evolution of minority retrieval performance during feedback-driven updates on the Wikipedia Movie Plots dataset ($d = 384$). The system exhibits a metastable regime ($t < 1500$) where minority recall (orange) remains high, masking the accumulation of geometric risk. As the minority embeddings silently drift toward the majority manifold (blue curve showing decreasing inter-cluster distance), they cross a critical interaction boundary. This triggers a sudden, irreversible collapse into a winner-take-all state where the agent exclusively serves the majority.

## 5. Related Work

**Retrieval-Augmented Generation (RAG).** RAG systems (Lewis et al., 2020; Izacard et al., 2021; Guu et al., 2020) have become a dominant paradigm for grounding LLMs in external knowledge. Numerous works have explored improving retrieval accuracy (Karpukhin et al., 2020; Ni et al., 2022), hybrid retrievers (Ram et al., 2023), scalable vector search (Douze et al., 2025; Rusum & Anasuri, 2024), and reranking (Nogueira & Cho, 2019; Ma et al., 2021). However, nearly all evaluation methodologies treat queries as i.i.d. and independent across users. Our work highlights that this assumption overlooks the system-level interactions induced by shared embedding spaces, revealing population-dependent failure modes absent from per-query evaluations.

**Embedding Geometry, Crowding, and Hubness.** High-dimensional embeddings suffer from well-known geometric pathologies, including concentration of measure (Vershynin, 2018), hubness (Radovanovic et al., 2010; Tomasev et al., 2013), anisotropy (Gao et al., 2019), and manifold collapse (Wang & Isola, 2020). These phenomena degrade nearest-neighbor retrieval quality and disproportionately affect low-density regions. We show that these representational limits operate as a *population-level mechanism* for exclusion: the interaction of these geometric constraints with heterogeneous user distributions leads to systemic marginalization of minority goals.

**Fairness and Bias in Information Retrieval.** Fairness in ranking and retrieval systems (Wang et al., 2023; Beutel et al., 2019; Celis et al., 2017; Dai et al., 2024; Hu et al., 2024; Kim & Diaz, 2025), and generative model bias (Bender et al., 2021; Qadri et al., 2025; Ghosh et al., 2024) have been studied extensively. Classical popularity bias often stems from historical interaction data or algorithmic amplification (Klimashevskaia et al., 2024). In contrast, our static model proves that a foundational form of popularity bias—the exclusion of minority intents—emerges endogenously purely from the spatial crowding of the shared embedding space, even before any interaction or dynamic feedback occurs. However, existing analyses predominantly treat retrieval as a sequence of isolated decisions, overlooking the feedback-driven evolution of the underlying embedding space. We extend this perspective to the geometric and dynamical domain and demonstrate that endogenous unfairness emerges naturally from goal collisions and density-dependent updates, even when the learning objective is accuracy-driven.

**Mean-Field Theory and Multi-Agent Learning.** Mean-field theory provides a rigorous framework for modeling

large interacting populations (Lasry & Lions, 2007; Carmona et al., 2018), with applications spanning neural networks (Mei et al., 2018; Sirignano & Spiliopoulos, 2020) and multi-agent reinforcement learning (Yang et al., 2018). We extend this formalism to retrieval systems and show that standard learning updates induce document embedding evolution governed by the non-linear Fokker–Planck equation. Our formulation aligns with foundational analyses of self-organization (Krichene et al., 2015; Chewi et al., 2020), yet uncovers a distinct winner-take-all phenomenon where density-amplified geometric forces drive the active subsumption of minority goals by the majority.

**Systemic Feedback and Representational Limits.** Recent work highlights systemic risks in AI, including self-preferencing (Shumailov et al., 2023), homogenization (Jiang et al., 2025), and feedback loops that drive rich-get-richer dynamics (Park et al., 2023; Liang et al., 2022; Morik et al., 2020; Singh & Joachims, 2018). Furthermore, prior works study bias amplification via synthetic data generation in self-consuming loops (Taori & Hashimoto, 2023; Wyllie et al., 2024). Our dynamic model proves a distinct, non-generative mechanism: marginalization driven strictly by gradient updates attempting to maximize local retrieval accuracy in a continuous, crowded space. While embedding models possess known expressivity bounds (Weller et al., 2025) and suffer from data skew (Bender et al., 2021), we identify a complementary, population-dependent failure mode. We show that shared embedding spaces constitute an implicit resource allocation system where geometric crowding and feedback dynamics drive minority collapse independently of model capacity or biased learning objectives.

## 6. Discussion and Conclusion

This paper presents a mean-field framework to analyze retrieval-augmented agents in multi-user settings. We demonstrate that shared embedding geometry induces emergent unfairness. We prove the existence of a phase transition in static retrieval where minority performance catastrophically collapses. Furthermore, we derive dynamic equations which reveal that standard feedback-driven learning drives the system to actively marginalize minority groups. These findings reframe RAG evaluation from isolated query metrics to collective multi-agent dynamics and expose critical failure modes for latent interests where metadata is absent.

**Deployability and Mitigation.** Our findings not only expose systemic risks but also suggest concrete deployment strategies for mitigating them. One effective approach is Metadata Pre-Filtering (Hard Subspace Isolation), where structured attributes (e.g., genre) are applied before vector similarity search. Restricting retrieval to a minority-specific subspace eliminates the influence of the surrounding major-

ity density, thereby avoiding the spatial crowding effect and preventing the drift-induced marginalization characterized in Theorem 3.1. In addition, the theoretical critical threshold $N_c$ provides a practical early-warning signal. By continuously estimating local document intensity and monitoring its proximity to $N_c$, system administrators can assess the geometric stability of the embedding space and trigger adaptive interventions before a sharp phase transition occurs.

**Vulnerabilities in Graph-Based ANN.** It is important to note that practical graph-based Approximate Nearest Neighbor (ANN) algorithms, such as HNSW, actually amplify rather than mitigate these geometric crowding effects. Because high-density majority documents act as statistical hubs, greedy graph routing paths become strongly biased toward these majority clusters. Consequently, a small local radius search around a minority query quickly saturates with majority interferers, causing the search to hit a local minimum and terminate early without finding the minority targets.

**Limitations.** Our theoretical framework relies on necessary abstractions. We assume a one-to-one query-document correspondence to derive clean analytic bounds; while multi-document relevance delays the phase transition slightly, it does not prevent the fundamental geometric collapse. Our dynamic model operates in the thermodynamic limit, assuming an effectively infinite document population, whereas real RAG systems operate at finite scales. Finally, while we propose metadata filtering as a safeguard, empirical testing of algorithmic mitigations—such as density-aware regularization—is left to future work.

## Impact Statement

This work demonstrates that geometric crowding in shared embedding spaces induces emergent unfairness in retrieval-augmented agents. We show that minority user goals undergo abrupt performance collapse even when faithfully reflecting population statistics. Our analysis reframes retrieval as a collective population dynamic and provides a theoretical foundation to diagnose systemic exclusion. These findings highlight a critical societal risk where standard accuracy optimization drives cultural homogenization and the erasure of specialized knowledge. We aim to guide the development of inclusive and robust retrieval-augmented agents.

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

# A. Proofs

In this appendix, we provide detailed mathematical derivations for the theorems presented in the main paper.

## A.1. Proof of Theorem 2.2: Minority Collapse via Shortlist Exclusion

*Proof.* Let $\mathbf{x} \in \mathcal{S}_{\min}$ be a query from a minority user. The retrieval system returns a shortlist of size $L$. The target document $d^*$ corresponds to query $\mathbf{x}$. A failure occurs if $L$ or more interfering documents from the majority class are closer to $\mathbf{x}$ than $d^*$ is.

We model the locations of the majority documents $\{\mathbf{d}_j\}_{j=1}^{N_{\mathrm{maj}}}$ as a realization of a non-homogeneous Poisson point process (PPP) with intensity function $\lambda(\mathbf{z}) = N_{\mathrm{maj}}\rho_{\mathrm{maj}}(\mathbf{z})$. Let $V(\mathbf{x})$ be the volume of the ball centered at $\mathbf{x}$ with radius $\|\mathbf{x} - \mathbf{d}^*\|$. The number of majority documents falling within this ball, denoted by the random variable $X_{\mathrm{maj}}$, follows a Poisson distribution:

$$X_{\mathrm{maj}} \sim \mathrm{Poisson}(\Lambda(\mathbf{x})),$$

where the rate parameter is the expected number of interferers:

$$\Lambda(\mathbf{x}) = \int_{V(\mathbf{x})} N_{\mathrm{maj}}\rho_{\mathrm{maj}}(\mathbf{z})d\mathbf{z} = N_{\mathrm{maj}} \cdot I(\mathbf{x})$$

Here, $I(\mathbf{x})$ represents the integral of the majority density over the interference volume. Recall that $I_{\min} := \inf_{\mathbf{x} \in \mathcal{S}_{\min}} I(\mathbf{x})$.

The target document is included in the shortlist if and only if fewer than $L$ interferers are present in the ball $V(\mathbf{x})$. Thus, the probability of inclusion is:

$$P(\text{Inclusion}) = P(X_{\mathrm{maj}} \leq L - 1) = \sum_{k=0}^{L-1} \frac{e^{-\Lambda(\mathbf{x})}(\Lambda(\mathbf{x}))^k}{k!} \tag{3}$$

We define the critical population threshold as $N_c := L/I_{\min}$. Thus for $N_{\mathrm{maj}} > N_c$, the expected number of interferers satisfies $\Lambda(\mathbf{x}) \geq N_{\mathrm{maj}}I_{\min} > L$ for all $\mathbf{x}$. Consider the regime where $\Lambda(\mathbf{x}) = (1 + \delta)L$ for some $\delta > 0$. Since we are in the regime where the mean $\Lambda > L$, we are interested in the probability of the random variable $X_{\mathrm{maj}}$ taking a value smaller than its mean (the left tail). We apply the Chernoff bound for the left tail of a Poisson distribution. For a Poisson variable $K$ with mean $\lambda$, and for any $k < \lambda$:

$$P(X \leq k) \leq \frac{e^{-\lambda}(e\lambda)^k}{k^k}$$

Substituting $\lambda = \Lambda(\mathbf{x})$ and $k = L - 1$:

$$P(X_{\mathrm{maj}} \leq L - 1) \leq \frac{e^{-\Lambda(\mathbf{x})} \cdot (e\Lambda(\mathbf{x}))^{L-1}}{(L-1)^{L-1}}$$

Taking the logarithm of the bound:

$$\ln P(\text{Inclusion}) \leq -\Lambda(\mathbf{x}) + (L-1)(1 + \ln \Lambda(\mathbf{x}) - \ln(L-1))$$

Substituting $\Lambda(\mathbf{x}) = N_{\mathrm{maj}}I(\mathbf{x})$:

$$\ln P(\text{Inclusion}) \leq -N_{\mathrm{maj}}I(\mathbf{x}) + L\ln(N_{\mathrm{maj}}) + c$$

where $c$ collects constant terms regarding $I(\mathbf{x})$ and $L$.

We now analyze the behavior as $N_{\mathrm{maj}} \to \infty$ to show asymptotic collapse. The dominant term in the exponent is $-N_{\mathrm{maj}}I(\mathbf{x})$. Since $I(\mathbf{x}) \geq I_{\min}$, we can bound this uniformly:

$$P(\text{Inclusion}) \leq \exp\left(-N_{\mathrm{maj}}I_{\min} + O(\ln N_{\mathrm{maj}})\right).$$

This confirms that the probability does not merely decay polynomially, but decays exponentially with respect to the population size of the majority. Consequently,

$$\lim_{N_{\mathrm{maj}} \to \infty} \mathbb{E}[P_{success}] = 0.$$

This completes the proof of the catastrophic collapse. $\square$

**A.2. Proof of Corollary 2.3: Multi-Document Relevance**

*Proof.* Let $\epsilon_1 \leq \epsilon_2 \leq \cdots \leq \epsilon_T$ be the distances from the query to these $T$ targets. To fail, the system must push all $T$ targets out of the shortlist. Because the targets are ordered by distance, if the closest target (at distance $\epsilon_1$) is excluded by $L$ interferers, all targets further away ($\epsilon_i > \epsilon_1$) are strictly excluded by those exact same $L$ interferers. Therefore, the probability of complete failure (excluding all $T$ targets) reduces entirely to the probability of excluding the nearest target.

Mathematically, increasing the number of ground-truth targets $T$ only shrinks the distance to the first-order statistic $\epsilon_1$, which marginally reduces the retrieval ball volume $V_d(\epsilon_1)$. This slightly increases the critical threshold $N_c$ and delays the start of the collapse. However, the dependence on $N_{maj}$ remains in the exponent. Once the new threshold $N_c$ is crossed, the probability of finding the nearest target (and thus any of the $T$ targets) decays exponentially, as bounded in Theorem 2.2. $\square$

**A.3. Proof of Theorem 3.1: Emergent Marginalization**

*Proof.* The steady state $\rho_D^*(\mathbf{x})$ minimizes the free energy functional $\mathcal{F}[\rho_D]$ subject to the mass conservation constraint $\int \rho_D(\mathbf{x})d\mathbf{x} = 1$. The first variation is given by:

$$\frac{\delta \mathcal{F}}{\delta \rho_D} = -CK\rho_G(\mathbf{x})e^{-C\rho_D(\mathbf{x})} + D(\ln \rho_D(\mathbf{x}) + 1) = \mu$$

where $\mu$ is the Lagrange multiplier. This steady-state minimizes the free energy functional $\mathcal{F}[\rho_D]$ by balancing two forces: the Interaction Potential (pulling documents toward frequently queried regions) and Entropic Regularization (spreading documents out). In the limit where the diffusion coefficient $D \to 0$ (or equivalently, the learning pressure $\beta = CK/D \to \infty$, representing the practical regime where accuracy-driven drift dominates random exploration), the entropic term $D(\ln \rho_D + 1)$ becomes negligible for strictly positive densities. The equilibrium is determined by the balance between the interaction potential and the Lagrange multiplier $\mu$:

$$-CK\rho_G(\mathbf{x})e^{-C\rho_D^*(\mathbf{x})} \approx \mu$$

Here, the left side represents the "attractive force" (the marginal retrieval gain), while the right side, $\mu$, represents the implicit system cost (the baseline representational cost). Solving for $\rho_D^*(\mathbf{x})$:

$$e^{-C\rho_D^*(\mathbf{x})} = \frac{-\mu}{CK\rho_G(\mathbf{x})} \implies -C\rho_D^*(\mathbf{x}) = \ln\left(\frac{-\mu}{CK\rho_G(\mathbf{x})}\right)$$

$$\rho_D^*(\mathbf{x}) = \frac{1}{C}\ln\left(\frac{CK\rho_G(\mathbf{x})}{-\mu}\right)$$

Let $\tau = \frac{-\mu}{CK}$. Since document density must be non-negative, $\rho_D^*(\mathbf{x})$ is defined only where $\rho_G(\mathbf{x}) > \tau$. In regions where the user goal density $\rho_G(\mathbf{x}) \leq \tau$, the attractive force is insufficient to support any document density against the implicit system costs (represented by $\mu$), clamping the solution to zero. Thus

$$\rho_D^*(\mathbf{x}) = \max\left(0, \frac{1}{C}\ln\left(\frac{\rho_G(\mathbf{x})}{\tau}\right)\right)$$

This confirms that minority interests below the critical density threshold $\tau$ suffer total representational collapse, i.e., $\rho_D^* = 0$.

$\square$

## B. Analytical Results for Gaussian Mixtures

We now derive exact, closed-form solutions for the minority success probability by instantiating the general framework of Theorem 2.2 with a Gaussian Mixture Model. These results quantify the phase transition, providing explicit formulae that allow for the direct calculation of critical collapse thresholds.

**Proposition B.1** (Success Probability in $d$-Dimensions)**.** *Consider a $d$-dimensional embedding space where the majority goals follow an isotropic multivariate Gaussian distribution centered at the origin, with variance $\sigma^2$ along each dimension. The density is given by:*

$$\rho_{maj}(\mathbf{x}) = \frac{1}{(2\pi\sigma^2)^{d/2}} \exp\left(-\frac{\|\mathbf{x}\|^2}{2\sigma^2}\right).$$

*Let $V_d(\epsilon)$ be the volume of the retrieval ball and $N_{maj}$ be the majority population size. Defining the interference constant $C_{maj} \equiv N_{maj}V_d(\epsilon)$, the success probability for a single minority goal located at position $\mathbf{x}_{min} \in \mathbb{R}^d$ has the following closed form:*

$$P_{success}(\mathbf{x}_{min}) = \exp\left(-\frac{C_{maj}}{(2\pi\sigma^2)^{d/2}} \exp\left(-\frac{\|\mathbf{x}_{min}\|^2}{2\sigma^2}\right)\right),$$

*where $\|\mathbf{x}_{min}\|$ is the Euclidean distance of the minority goal from the center of the majority cluster.*

Although high-dimensional spaces theoretically offer vast volume, empirical studies have shown that neural embeddings suffer from severe anisotropy or the *cone effect*, where representations collapse into a narrow sub-manifold (Gao et al., 2019). This effectively reduces the usable capacity of the space, which makes collision inevitable even in high dimensions. Our dynamic analysis complements the work of (Wang & Isola, 2020) by demonstrating that repeated cycles of training or fine-tuning of retrieval components from user feedback exacerbate this collapse. This generalized formula shows the crucial role that dimensionality plays in system performance. The fundamental double-exponential relationship between success probability and distance from the interfering cluster still holds. However, the new term $(2\pi\sigma^2)^{d/2}$ in the denominator grows exponentially with the dimension $d$. This implies that for a fixed number of users, their peak density at the center of the cluster is substantially lower in higher dimensions, as the embedding volume is much larger. This effect provides a *blessing of dimensionality* (Gorban et al., 2020) in this context, which reduces the severity of interference at the core of the majority cluster and provides a theoretical justification for the use of high-dimensional embeddings in practice.

To build intuition for the geometric mechanism underlying goal collision, we consider a minimal but analytically transparent setting, where the embedding space is one-dimensional ($d = 1$) and has a heterogeneous user population. This example captures the essential interaction between population density, geometric crowding, and retrieval performance. Assume that user goals are drawn from a mixture of two Gaussian distributions,

$$\rho_G(\mathbf{x}) = (1 - \alpha)\,\mathcal{N}(\mathbf{x} \mid \mu_1, \sigma_1^2) + \alpha\,\mathcal{N}(\mathbf{x} \mid \mu_2, \sigma_2^2),$$

where $\alpha \ll 0.5$ denotes the fraction of minority users. The majority population is concentrated around $\mu_1$ with small variance $\sigma_1^2$, yielding a high local density, and the minority population is centered at $\mu_2$ with larger variance $\sigma_2^2$, corresponding to a diffuse, low-density region of the embedding space. We assume that $|\mu_1 - \mu_2|$ is large enough that the two modes are initially well separated.

Consider first a majority user whose query embedding lies near the mode $\mathbf{x} = \mu_1$. The local goal density $\rho_G(\mu_1)$ is high, and by Eq. (1), the success probability

$$P_{\text{success}}(\mu_1) \approx \exp(-M\rho_G(\mu_1)V_1(\epsilon)),$$

may be substantially less than one due to frequent collisions. However, these collisions predominantly involve documents associated with *other majority goals* that are nearby in the embedding space. As a result, although strict nearest-neighbor correctness may degrade, the retrieved documents often remain semantically aligned with the majority user's broad intent. In this sense, majority users are relatively robust to geometric interference. In contrast, consider a minority user with a query near $\mathbf{x} = \mu_2$. The intrinsic density contributed by the minority population, $\alpha\mathcal{N}(\mu_2 \mid \mu_2, \sigma_2^2)$, is low. However, the success probability at $\mu_2$ is not determined solely by this local minority density. Instead, it is governed by the *total* goal density $\rho_G(\mu_2)$, which includes contributions from the tails of the majority distribution. Even when $\mu_2$ lies far from $\mu_1$, the exponentially decaying but non-zero tail of the majority Gaussian introduces interfering documents that compete with the minority target.

The mean-field perspective reveals an important systemic vulnerability. As the majority population grows, equivalently, as $(1 - \alpha)M$ increases, the peak density $\rho_G(\mu_1)$ increases proportionally. To maintain acceptable retrieval performance in this high-density region, the embedding space must effectively resolve documents at increasingly smaller scales, corresponding to a reduction in the characteristic separation $\epsilon$ between targets and competitors. It is important to note that this constraint is *global*, imposed by the densest region of the space and applies uniformly to all goals. For the minority population, whose typical inter-goal separation is governed by the larger variance $\sigma_2^2$, this global reduction in $\epsilon$ is catastrophic. The scale required to avoid collisions near $\mu_1$ becomes smaller than the natural spacing between minority targets and their nearest competitors. Consequently, the success probability at $\mu_2$ collapses sharply,

$$P_{\text{success}}(\mu_2) \approx \exp(-M\rho_G(\mu_2)V_1(\epsilon)) \;\to\; 0,$$

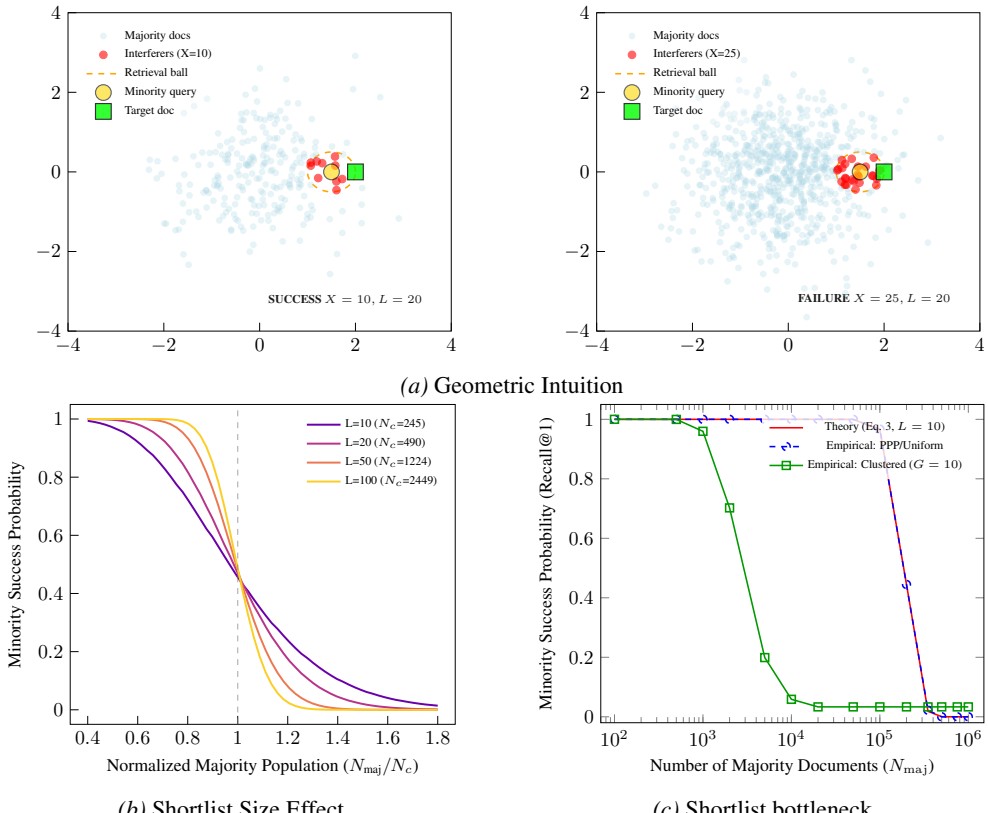

*(a)* Geometric Intuition

*(b)* Shortlist Size Effect                    *(c)* Shortlist bottleneck

*Figure 6.* **Geometric Intuition and Clustering Effects.** (a) Visualization of goal collision: As majority density increases, the retrieval ball around a minority query fills with interfering documents that displace the target. (b) Increasing the shortlist budget $L$ sharpens the phase transition but does not prevent it. (c) Impact of clustering: Comparison of the theoretical PPP baseline (blue) against a realistic clustered Gaussian Mixture (green). Clustering creates pockets of super-critical density that cause the system to collapse at significantly lower population sizes $N_{\text{maj}}$, than uniform density models predict. This result confirms that our theoretical bounds are conservative.

even though neither the minority population size nor its internal structure has changed.

This example highlights the insight from our analysis that retrieval performance for minority goals can deteriorate dramatically due to growth in an *unrelated* majority population. The failure is not driven by changes local to the minority cluster, but by geometric pressure exerted elsewhere in the embedding space. The two-population Gaussian mixture example illustrates how goal collision induces population-dependent phase transitions in retrieval success.

### B.1. Sensitivity to Point Process Assumptions

To address potential deviations from the PPP assumption, which implies a uniform background density, we compare our baseline model against a Clustered Gaussian Mixture Model ($G = 10$) with fixed dimensionality ($d = 64$) and shortlist size ($L = 10$). The PPP baseline models a homogeneous point process with constant interference probability of $5 \times 10^{-5}$ per document. For the clustered model, each cluster is designated as near with probability $0.3$ (interference probability $0.012$) or far with probability $0.7$ (interference probability $10^{-7}$), reflecting heterogeneous spatial density around the minority query. The results, visualized in Figure 6c, demonstrate that clustering accelerates the retrieval collapse, with the critical population threshold dropping from $N_{\text{maj}} \approx 200,000$ in the PPP baseline to $N_{\text{maj}} \approx 5,000$ in the clustered regime. This acceleration indicates that retrieval failure is driven by the peak local density of specific clusters rather than the global average density, causing minority queries near these clusters to face crowding even when the total system population is relatively small. This confirms that the PPP assumption in Theorem 2.2 is in fact a conservative lower bound; in practice, real-world clustering reduces the effective capacity of the embedding space and amplifies the marginalization mechanism.

## C. Derivation of the Interaction Potential

In the main paper, we introduced the interaction potential $\mathcal{V}[\rho_D] = \int \rho_G(\mathbf{x})e^{-C\rho_D(\mathbf{x})}dx$ as the driver of the drift dynamics. Here, we show that this functional is not an ad-hoc modeling choice, but rather the thermodynamic limit of a standard microscopic retrieval objective, maximizing system-wide recall (or equivalently, minimizing the retrieval failure rate).

Consider a discrete retrieval system consisting of $M$ document embeddings $\mathcal{D}_M = \{d_1, \ldots, d_M\} \subset \mathbb{R}^d$. For a given user query $q \in \mathbb{R}^d$ drawn from the goal distribution $\rho_G$, a retrieval event is considered successful if at least one document falls within a retrieval radius $\epsilon$ of the query. Conversely, a retrieval failure occurs if the query falls into a void where no documents are present. We define the microscopic loss function $\ell(q, \mathcal{D}_M)$ as the binary indicator of retrieval failure:

$$\ell(q, \mathcal{D}_M) = \mathbb{I}\left(\bigcap_{j=1}^{M}\{\|d_j - q\| > \epsilon\}\right).$$

The total system risk $\mathcal{R}_M$ is the expected failure rate averaged over the population of user goals:

$$\mathcal{R}_M = \mathbb{E}_{q \sim \rho_G}\left[\mathbb{E}_{\mathcal{D}_M}[\ell(q, \mathcal{D}_M)]\right].$$

We consider the standard mean-field scaling limit to derive the macroscopic field equation. We assume the document embeddings are exchangeable and become asymptotically independent as $M \to \infty$, distributed according to the smooth density $\rho_D(\mathbf{x})$. The probability that a single specific document $d_j$ falls within the retrieval ball $B_\epsilon(q)$ is given by the mass of the ball

$$p_\epsilon(q) = \int_{B_\epsilon(q)} \rho_D(z)dz \approx \text{Vol}(B_\epsilon)\rho_D(q).$$

The probability that all $M$ documents fail to match the query (the probability of a void) is therefore:

$$P(\text{Void}|q) = (1 - p_\epsilon(q))^M = (1 - \text{Vol}(B_\epsilon)\rho_D(q))^M.$$

We now take the limit where the number of documents $M \to \infty$ and the retrieval volume $\text{Vol}(B_\epsilon) \to 0$ such that the effective system capacity $C \equiv M \cdot \text{Vol}(B_\epsilon)$ remains constant. This scaling reflects a dense retrieval regime where the index size grows while the required precision sharpens. We can write

$$\lim_{M \to \infty}\left(1 - \frac{C\rho_D(q)}{M}\right)^M = e^{-C\rho_D(q)}.$$

Substituting the asymptotic void probability back into the system risk, we obtain the continuous loss functional:

$$\mathcal{R}[\rho_D] = \int_{\mathcal{S}} \rho_G(q)e^{-C\rho_D(q)}dq.$$

This functional is identical (up to constants) to the interaction potential $\mathcal{V}[\rho_D]$.

## D. Extended Discussion on Hubness and Dimensionality

In the main paper, we alluded to the role of geometric pathologies in high-dimensional embedding spaces. Here we elaborate on how hubness and anisotropy interact with the mean-field mechanisms studied in this paper and contribute to population-dependent retrieval collapse in RAG systems.

High-dimensional spaces are often assumed to be sparse; however, empirical studies of neural embeddings show marked deviations from this intuition. Nearest-neighbor relations in such spaces exhibit pronounced *hubness*, in which a small number of points appear disproportionately often in neighbor lists (Tomasev et al., 2013; Radovanovic et al., 2010). Additionally, transformer-based embeddings are known to be highly anisotropic, with representations concentrated in narrow regions of the ambient space (Ethayarajh, 2019; Godey et al., 2024). Both effects effectively increase local density and exacerbate geometric crowding. From a geometric perspective, hubness emerges alongside concentration of measure. As the embedding dimension $d$ increases, the contrast between pairwise distances diminishes. At the same time, the distribution of

$k$-occurrences, the number of times a point appears in the $k$-nearest-neighbor lists of other points, becomes highly skewed. A small number of points act as hubs that dominate retrieval neighborhoods, while the majority of points become *anti-hubs* and are rarely, if ever, retrieved. Our mean-field formulation provides a continuous approximation of this otherwise discrete phenomenon. In particular:

- High-density majority regions in the goal distribution act as statistical hubs in the embedding space.

- Documents associated with minority goals effectively become anti-hubs as their neighborhoods are increasingly populated by majority interferers.

- Equation (1), $P_{\text{success}} \approx \exp(-M\rho V)$, can be interpreted as governing a document's effective hubness: documents in high-density regions experience an exponentially reduced probability of being the nearest neighbor to a specific query due to geometric crowding within the retrieval volume $V$.

This suggests that fairness in RAG is not just about training data bias. Even if the training data is perfectly balanced, the geometry of the inference time embedding space will induce unfairness via hubness. Popular topics will geometrically dominate the retrieval lists, pushing niche topics out of the retrieval shortlist used by RAG systems.

## E. Connection Between Mean-Field Dynamics and Practical Learning Algorithms

In this appendix, we demonstrate how the population-level dynamics in Section 3 emerge directly from standard embedding update rules used in retrieval systems.

Let $d \in \mathbb{R}^d$ denote the embedding of a single document, and let $q \in \mathbb{R}^d$ denote a query embedding sampled from the goal distribution $\rho_G$. During learning, document embeddings are updated via stochastic gradient-based rules of the form

$$\Delta d \equiv d_{t+1} - d_t \propto -\nabla_d \ell(d, q; \rho_D) + \xi_t, \tag{4}$$

where $\ell(d, q; \rho_D)$ is a retrieval loss that depends on the document population (e.g., via contrastive normalization or negative sampling), and $\xi_t$ denotes stochastic noise originating from minibatching, sampling, or optimizer randomness. The negative gradient term $-\nabla_d \ell$ induces a drift of document embeddings toward regions of the embedding space frequently queried by users, while the noise term induces diffusion. Importantly, the query $q$ is drawn from the population-level goal distribution $\rho_G$, which is generally skewed. In the limit of small step sizes, frequent updates, and a large number of documents, the empirical distribution of document embeddings converges to a continuous density $\rho_D(\mathbf{x}, t)$ whose evolution is governed by a nonlinear drift–diffusion (McKean–Vlasov) equation. The expected drift $\mathbb{E}_{q \sim \rho_G}[-\nabla_d \ell(d, q; \rho_D)]$ recovers the negative gradient of the interaction potential $-\nabla(\delta \mathcal{V}/\delta \rho_D)$ defined in the macroscopic model. This yields Eq. (2), where the drift field corresponds to the expected gradient-induced motion modulated by competitive density effects, and the diffusion term captures the aggregate effect of stochasticity in the updates. The emergent behavior results solely from the interaction between population skew in $\rho_G$, shared embedding geometry, and feedback-driven learning.

The continuum approximation becomes appropriate when the number of documents is large and embedding updates are frequent. In the early stages of a system's lifecycle (small index, sparse embeddings), discrete stochastic effects and initialization noise dominate. However, as retrieval-augmented agents are deployed to a continuous user base and the index scales alongside high-frequency feedback loops, the system enters the thermodynamic limit. Empirically, our experiments suggest this macroscopic regime becomes highly relevant at relatively modest deployment scales. We observe density-induced collapse beginning at index sizes as small as $N_{maj} \approx 2000$ to $25000$ depending on the modality, and dynamic marginalization taking over after roughly $t \approx 1500$ simulated update steps. It is precisely in this mature, continuously updating regime that the macroscopic drift-diffusion dynamics—governed by the non-linear Fokker-Planck equation—become irreversible.

