# OpenReview forum: "The Crowded Embedding Space: A Mean-Field Mechanism for Emergent Marginalization in Retrieval-Augmented Agents"
_ICML.cc/2026/Conference — ICML 2026 regular_

### Official Review · Reviewer_vRrx · 2026-02-22

**Soundness:** 4
**Presentation:** 4
**Significance:** 4
**Originality:** 4
**Overall Recommendation:** 6
**Confidence:** 5

**Summary:**

The paper addresses the problem of semantic ambiguity in embeddings where intervening high frequency classes may disrupt the retrieval of rarer modes. The papers shows through a simple mean field approximation propagated through time as a focker planck drift-diffusion that standard training procedures force a phase transition beyond which this phenomena becomes irreversible.

**Compliance With Llm Reviewing Policy:**

Affirmed.

**Final Justification:**

My strong rating originates as a consequence of a relevance of this problem in practical setups especially enterprise settings where embedding based pipelines are central to processes. Furthermore, the authors have provided a clean framework using an intuitive approach from mean field theory (which is relevant for most practical scenarios as embeddings are mainly utilized in the large cardinality limit) with a clear description of a phase transition and its consequences. Hopefully this work can lead to novel insights that could be adopted by practitioners.

**Key Questions For Authors:**

I have no other questions and would like to thank the authors for their novel work!

**Limitations:**

Yes

**Strengths And Weaknesses:**

Strengths

1 The approach is genuinely novel and addresses an important problem that is central to practical setups but is not discussed in academic settings.
2 The logic chain is clear and the theory is self consistent and is well supported by extensive experiments.
3 The phase transition proposed can be easily empirically verified and could be utilized for evaluation of retrieval corpuses under continuous updation.

Weaknesses

1 Perhaps more attention could be given to the phase transition and its consequences related to how practical systems could adapt using this procedure.
2 The algorithm assumes nearest neighbors retrieval through direct search which is known to suffer from the curse of dimensionality, often graph based constructions such as HNSW with a small local radius search are used.
3 It would be interesting to see where the mean field approximation becomes relevant as embeddings are learnt over data sizes through time.

---

> ### Author Rebuttal · Authors · 2026-03-29
>
> We thank you for your strong support of our paper and are thrilled that you found the formalization compelling.
> Below, we address your suggestions for improving the completeness of the manuscript.
>
> **W1 Practical System Adaptation**
> We completely agree that our theoretical findings should translate into actionable strategies for system administrators. In the manuscript, we will expand the Conclusion to detail how the phase transition can serve as an early warning system. By dynamically estimating the local document intensity function and comparing it against the theoretical critical threshold $N_c$, systems can actively monitor the geometric health of their embedding space. If the majority density approaches this critical threshold, the system could automatically trigger adaptive interventions. Some of such interventions might include partitioning the vector index by metadata, dynamically expanding the shortlist budget before the strict phase transition is crossed, or allocating dedicated embedding subspaces to protect marginalized topics from catastrophic collapse.
>
> **W2 Graph-Based ANN Search and HNSW**
> You are absolutely right that our formal geometric proofs assume an exact nearest-neighbor search. However, in the revised manuscript, we will explicitly clarify that graph-based ANN algorithms actually amplify the geometric crowding effects we describe, rather than mitigating them. HNSW relies on greedy graph routing based on spatial proximity in the ambient space. Because high-density majority documents act as statistical hubs, the routing paths become strongly biased toward these majority clusters. Consequently, a small local radius search around a minority query will quickly become saturated with majority interferers. As a result, the ANN search hits a local minimum and terminates early and completely misses the minority targets. We touched on this briefly when discussing how restricted shortlist budgets in two-stage pipelines create a hard bottleneck, but we will add a dedicated paragraph connecting our theory directly to the mechanics and vulnerabilities of HNSW.
>
> **W3 Relevance of Mean Field Approximation over Time**
> As noted in Section 3.1, the continuum approximation becomes appropriate when the number of documents is large and when embedding updates are small but frequent. In the early stages of a system's lifecycle (small index, sparse embeddings), discrete stochastic effects and initialization noise dominate. However, as retrieval-augmented agents are deployed to a continuous user base and the index scales up alongside high-frequency feedback loops, the system enters the thermodynamic limit. Empirically, our experiments suggest this macroscopic regime becomes highly relevant at relatively modest deployment scales. For instance, we observed density-induced collapse beginning at index sizes as small as $N_{maj} \approx 2000$ to $25000$ depending on the modality, and dynamic marginalization taking over after roughly $t \approx 1500$ simulated update steps. It is precisely in this mature, continuously updating regime that the macroscopic drift-diffusion dynamics- governed by the non-linear Fokker-Planck equation - take over. We will add a brief discussion to Appendix E to explicitly state these scaling boundaries and clarify when the emergent marginalization becomes irreversible.

---

> > ### Author Rebuttal · Reviewer_vRrx · 2026-04-01
> >
> > a) I believe that the authors have adequately addressed my concerns regarding the aspects that I raised for their work.

---

### Official Review · Reviewer_oih7 · 2026-03-12

**Soundness:** 2
**Presentation:** 1
**Significance:** 2
**Originality:** 2
**Overall Recommendation:** 2
**Confidence:** 4

**Summary:**

The paper argues that retrieval-augmented generation (RAG) agents suffer from a systemic failure mode called "goal collision," where high document density for majority interests geometrically displaces minority content from retrieval results. Using mean-field theory and a  Fokker-Planck equation, the authors present a static and dynamic analysis showing that as majority data scales, minority user goals undergo a sharp, catastrophic "phase transition" in performance. Then in experiments on text, vision, the authors show this collapse. They also indicate that increasing the retrieval budget or using rerankers fails to mitigate the issue.

**Compliance With Llm Reviewing Policy:**

Affirmed.

**Key Questions For Authors:**

- The analysis relies on modeling document locations as a Poisson Point Process (PPP) does Theorem 2.2 holds for other document distributions?
- Can you discuss the proof for Theorem 3.1 in more detail?

**Limitations:**

Yes

**Strengths And Weaknesses:**

Strengths
- They connect retrieval to mean-field theory/stochastic geometry to model population-level behavior
- The experiments validate their phase transition theory on text (QQP) and vision (CIFAR-100), and narrative data (Wikimedia Movie Plots)

Weaknesses
- The concept of a "user goal" remains abstract and somewhat unclear in the context of a functional RAG system ; the paper simplifies the complex goal distribution by modeling it as a binary mixture of a dominant majority and a sparse minority
- The analysis relies on modeling document locations as a Poisson Point Process (PPP) ; it is unclear if Theorem 2.2 holds for other document distributions
- The proof for Theorem 3.1 resembles an partial differential equation (PDE) solution and several terms are thrown in without sufficient derivation or discussion. Some examples: "the learning pressure" β = CK/D → ∞, "attractive force is insufficient to support any document density against the implicit system costs".
- Their model assumes a one-to-one correspondence where each user goal $g_i$ has exactly one correct target document $d_i$ with initially close embeddings, this does not capture the semantic ambiguity of real-world retrieval.
- The practical implications for deployed systems remain unclear, and the paper fails to clearly articulate how its specific embedding dynamics in Section 3.1 relate to the findings in the cited literature, such as Vu et al. or Mignacco et al.

---

> ### Author Rebuttal · Authors · 2026-03-30
>
> We appreciate your valuable feedback and recognition of our contributions. To clarify our dynamic model’s physical intuition, we will insert a bridging paragraph into Section 3 to map our formal derivations directly to practical RAG mechanics. We address your specific concerns below, and promote essential analyses from the appendices to the main text to ensure our core arguments are transparent and clear.
>
> **W1 Abstraction of User Goals and Binary Mixtures**
> While Section 2.1 uses a binary mixture to build intuition, our theoretical framework is not restricted to it. Theorem 3.1 explicitly models $\rho_G(x)$ as an arbitrary continuous distribution with multiple disjoint local maxima to capture heterogeneous, multi-topic user populations. The binary case cleanly isolates geometric interference effects, but the emergent marginalization mechanism generalizes to complex distributions.
>
> **W4 Semantic Ambiguity**
> We agree that real-world retrieval is semantically ambiguous, but the 1:1 mapping in our static theory was chosen as a deliberate, minimal analytic assumption. To rigorously prove the existence of the geometric interference mechanism, we needed a clean, binary definition of "retrieval success." Introducing soft relevance or multi-document targets into the foundational proof would needlessly complicate the math without qualitatively altering the geometric realities we describe. However, we do not rely on this 1:1 assumption in practice. Our empirical evaluations show that these geometric limits apply directly to messy, ambiguous real-world data. For example, our experiment on the Quora Question Pairs (QQP) dataset (Section 4.3, Figure 4d) explicitly models semantic interference. Despite the high semantic overlap in QQP, the data exhibits the exact same phase transition and catastrophic minority collapse predicted by our 1:1 theoretical model.
>
> **W5 Practical Implications & Cited Literature**
> Deployments increasingly use continuous index refreshing. Vu et al. (2024) show RAG systems update indices via feedback, and Mignacco et al. (2020) prove these updates behave as mean-field drift-diffusion. Bridging these, we reveal that standard accuracy-maximizing updates mathematically and irreversibly erase minority knowledge. To mitigate this, we propose Metadata Pre-filtering. As detailed to Reviewer BAo6, filtering on structured tags (e.g., genre) prior to search constrains queries to an isolated subspace which effectively zeros out majority density. This prevents the drift-induced marginalization proven in Theorem 3.1. We will add a "Deployability" paragraph to Section 5.
>
> **W2/Q1 Does Theorem 2.2 hold for other distributions?**
> Yes, and the collapse actually becomes more severe. We analyzed this in Appendix B.1 and Figure 6c. The Poisson Point Process (PPP) models a homogeneous background, serving as a conservative lower bound. Real-world distributions are clustered. Our Clustered Gaussian Mixture Model shows that clustering accelerates the performance collapse, dropping the critical majority threshold from $N_{maj} \approx 200,000$ (PPP) to $N_{maj} \approx 5,000$ (clustered). We will move a summary of Appendix B.1 to the main text to clarify this.
>
> **W3/Q2 Discussion of Theorem 3.1 proof in detail**
> The parameters comprising the "learning pressure" limit ($\beta = CK/D \rightarrow \infty$) were defined in Section 3.1. However, we agree that the proof in Appendix A.2 would benefit from directly mapping these mathematical terms to the physical intuition and providing more details. We will expand the proof as follows:
>
> Theorem 3.1 analyzes the steady-state of the McKean-Vlasov equation (Eq. 2), setting $\partial\rho_D/\partial t = 0$. The steady-state minimizes the free energy functional $\mathcal{F}[\rho_D]$, which balances two forces:
> - Interaction Potential (Drift): Pulls documents toward frequently queried regions, weighted by learning rate $K$.
> - Entropic Regularization (Diffusion): Spreads documents out, weighted by noise $D$. The term $\beta = CK/D$ is the effective "learning pressure" (the ratio of exploitation to exploration). When we state $\beta \rightarrow \infty$, we refer to the practical regime where accuracy-driven updates (drift) dominate random exploration (diffusion). Solving the variational derivative $\frac{\delta\mathcal{F}}{\delta\rho_D} = \mu$ (where $\mu$ is the Lagrange multiplier for mass conservation) directly yields the equilibrium state: $-CK\rho_G(x)e^{-C\rho_D^*(x)} \approx \mu$
>
> The left side is the "attractive force" (marginal retrieval gain). The right side, $\mu$, is the implicit system cost (baseline representational cost).
>
> Solving for $\rho_D^*(x)$ yields logarithmic dependence on $\rho_G(x)$. If goal density $\rho_G(x)$ is too low to overcome cost $\mu$ (falling below threshold $\tau = -\mu/CK$), the equation implies negative density. Since physical density must be non-negative, the system clamps it to zero. We will add this to App. A.2.

---

> > ### Author Rebuttal · Reviewer_oih7 · 2026-04-07
> >
> > Thank you for the clarifications. However, my original evaluation still stands, and the paper would benefit from a substantial rewrite

---

### Official Review · Reviewer_BAo6 · 2026-03-12

**Soundness:** 3
**Presentation:** 3
**Significance:** 2
**Originality:** 2
**Overall Recommendation:** 3
**Confidence:** 3

**Summary:**

This paper shows that shared embedding spaces in RAG can crowd out minority intents, leading to systematic retrieval failures as majority content becomes dense.

**Compliance With Llm Reviewing Policy:**

Affirmed.

**Final Justification:**

The response addresses part of my concerns, and I am willing to raise my soundness score. However, I am keeping my overall recommendation at 3 (Weak Reject), because the main issues still remain, especially the limited discussion of closely related work, the reliance on strong assumptions. Addressing these concerns would likely require revisions to the paper.

**Key Questions For Authors:**

Please refer to weaknesses, especially:
1. How do results change with multi-document relevance per query?
2. Have you considered the impact of any mitigation measures that could be implemented in practice?

**Limitations:**

yes

**Strengths And Weaknesses:**

***Strengths***
- This is an important problem. The core observation is clear: dense regions can crowd out minority queries.
- The paper is well-structured and easy to follow.
- The validation covers settings (text, vision, style, and topic).

***Weaknesses***
- Related work is under-discussed:
    - The phenomenon is closely connected to popularity bias[1] in recommender and retrieval systems, including known mitigation strategies, but the paper does not discuss these results.
    - Model collapse in self-consuming models has been mentioned, but a closer connection is bias amplification[2,3] in self-consuming loops.
- The theory relies on some strong assumptions:
    - One-to-one query-document correspondence
    - $k=1$ nearest neighbor
    - The assumption $\lambda_D(x) \approx M\rho_G(x)$: this is especially unclear in the static setting, where a fixed corpus (and embedding space) should not track user demand.
- The paper does not evaluate any mitigation strategies, which limits its practical impact.

[1] Klimashevskaia, A., Jannach, D., Elahi, M. et al. A survey on popularity bias in recommender systems. User Model User-Adap Inter 34, 1777–1834 (2024). https://doi.org/10.1007/s11257-024-09406-0

[2] Wyllie, S., Shumailov, I., and Papernot, N. Fairness feedback loops: Training on synthetic data amplifies bias. In Proceedings of the 2024 ACM Conference on Fairness, Accountability, and Transparency, FAccT’24, pp.2113–2147, New York, NY, USA, 2024.

[3] Taori, R. and Hashimoto, T. B. Data feedback loops: Model-driven amplification of dataset biases. In Proceedings of the 40th International Conference on Machine Learning, ICML’23. JMLR.org, 2023.

---

> ### Author Rebuttal · Authors · 2026-03-29
>
> We thank the reviewer for taking the time to read our work and provide valuable feedback, and for recognizing the importance of the problem, the clarity of our core observation, and presentation. We address your specific concerns below.
>
> **W1 Related Work: Popularity Bias & Bias Amplification**
> Thanks for highlighting these connections which strongly contextualize our contributions. We will integrate them into Section 5 (Related Work):
> * **Popularity Bias [1]:** Classical popularity bias often stems from historical interaction data or algorithmic amplification. However, our static model (Section 2) proves that a foundational form of popularity bias (the exclusion of minority intents) emerges *endogenously* purely from the spatial crowding of the shared embedding space, even before any interaction or dynamic feedback occurs.
> * **Bias Amplification [2, 3]:** We will include Taori & Hashimoto (2023) and Wyllie et al. (2024) alongside our discussion of model autophagy. These works study bias amplification via synthetic data generation in self-consuming loops, whereas our dynamic model (Theorem 3.1) proves a distinct, non-generative mechanism; That is marginalization driven strictly by gradient updates attempting to maximize local retrieval accuracy in a continuous, crowded space.
>
> **W2 Strong Assumptions: Nearest Neighbor & 1-to-1 Correspondence**
> We used 1-to-1 correspondence and $k=1$ nearest neighbor strictly as a minimal baseline in Section 2.1 to derive clean, interpretable bounds. However, our core phenomena hold and often worsen under relaxed assumptions.
> Our primary theoretical result, Theorem 2.2, explicitly relaxes the $k=1$ assumption by modeling a two-stage retrieval pipeline with a shortlist of size $L$. Moreover, our experiments empirically evaluate Recall@$k$ for $k$ in {$1, 2, 5, 10$} (Figure 4). The results confirm that expanding the retrieval budget provides negligible mitigation against density-induced exclusion; it simply delays the phase transition slightly before geometric crowding overwhelms the shortlist. (For the 1-to-1 assumption, please see Q1 below).
>
> **W2 The Static Density Assumption**
> The reviewer rightly points out that a strictly static index cannot actively track user demand, i.e., $\lambda_D(x) \approx M\rho_G(x)$. We will clarify in Section 2.1 that, in real-world static corpora (e.g., Wikipedia, the Web, or corporate knowledge bases), the supply of documents naturally mirrors historical human interest. Mainstream topics organically accumulate larger volumes of written content than niche topics. Therefore, even a fixed snapshot of a corpus inherently possesses the skewed density distribution that triggers the geometric interference we describe.
>
> **W3 Lack of Mitigation Strategies**
> While the primary contribution of this paper is foundational - to formally diagnose and mathematically bound this geometric failure mode - we agree that discussing actionable mitigations adds significant practical impact. We detail an effective mitigation strategy below in our response to Q2, which we will add to our Discussion section.
>
> **Q1 Multi-document Relevance Per Query**
> Relaxing the one-to-one correspondence assumption to require multi-document relevance actually *amplifies* the geometric vulnerability. If a query requires $m$ relevant documents to succeed (where $m > 1$), the retrieval neighborhood must be large enough to encompass $m$ minority targets before becoming saturated by majority interferers. Because the success probability decays exponentially with the density of the majority (Equation 1), requiring multiple targets accelerates the phase transition, causing the minority collapse to happen at significantly lower majority population thresholds. We will formalize this as a corollary in Appendix A.
>
> **Q2 Impact of Mitigation Measures in Practice**
> Based on our theoretical framework, we can confirm a practical mitigation strategy, namely Metadata Pre-Filtering (Hard Subspace Isolation), which we will detail in the paper.
> As noted in our conclusion, geometric collapse is particularly critical for latent interests where metadata is absent. However, if structured metadata is available (e.g., tagging documents with explicit "Film Noir" vs. "Crime/Mystery" genre labels), RAG systems can apply a deterministic metadata filter before executing the vector similarity search. By doing so, the query is evaluated exclusively within the isolated minority subspace. This effectively zeroes out the majority density within the retrieval volume, completely bypasses the spatial crowding problem, and guarantees protection from the geometric interference mechanism we describe.

---

> > ### Author Rebuttal · Reviewer_BAo6 · 2026-04-03
> >
> > Thanks for the rebuttal. It is stated that mainstream topics have “larger volumes” of content, but this is a corpus-wide argument, not a statement about local density in embedding space. What matters here is the local intensity $\lambda_D(x)$ around a query. Is it possible that documents for mainstream topics are diffuse, while documents for niche topics form tight clusters?

---

> > > ### Author Response · Authors · 2026-04-03
> > >
> > > Thanks for the insightful follow up. It is correct that local intensity $\lambda_D(x)$ (rather than global corpus volume) governs the retrieval collision mechanics. Your intuition regarding a highly diffuse majority and a tightly clustered minority is spot-on in that a tighter minority shrinks the retrieval ball $V(x)$ which effectively delays the phase transition threshold $N_c$. However, our framework accounts for this topology both theoretically and empirically, and shows that marginalization still inevitably occurs. We will clarify this in the final manuscript as follows:
> > >
> > > 1. Even if the majority distribution is highly diffuse and the minority is tightly clustered, the phase transition still occurs and it is only delayed but not prevented. In our model, the local intensity of interferers is $\lambda(x) = N_{maj}\rho_{maj}(x)$. As detailed in Proposition B.1, our closed-form solution for Gaussian mixtures already explicitly models this variance ($\sigma^2$) dynamic. Even if the minority cluster is tight and located far from the majority mean, it still resides within the exponentially decaying tail of the majority distribution. Because the local intensity $\lambda(x)$ scales linearly with $N_{maj}$, an increasing diffuse majority population will continuously raise the absolute background density across the entire space. Eventually, the tail density $\Lambda(x)$ intersecting the minority neighborhood will cross the critical shortlist threshold $\Lambda(x) > L$ which triggers the exact same super-polynomial collapse described in Theorem 2.2.
> > >
> > > 2. A perfectly diffuse majority is possible in an ideal isotropic space, however, as referenced in the manuscript real-world LLM and vision embeddings suffer from severe anisotropy and hubness (as detailed in Appendix D, references Gao et al., 2019; Ethayarajh, 2019). Neural representations tend to collapse into narrow sub-manifolds. As a result, the usable capacity of the space is reduced, which forces mainstream topics to form locally dense statistical hubs rather than spreading out diffusely. So, the geometric constraints of the embedding models themselves enforce the high local density $\lambda_D(x)$ that drives our interference mechanism.
> > >
> > > 3. We validated the phase transition on real-world datasets (e.g. QQP, CIFAR-100, 20 Newsgroups) to ensure that our claims hold beyond idealized density assumptions. These datasets contain natural variations in topic diffuseness and clustering, and yet, we observed the predicted catastrophic collapse universally across all modalities, despite these topological irregularities (Figure 4).
> > >
> > > We will add a discussion section in the body of the manuscript explicitly discussing the diffuse majority vs tight minority variance scenario and clarify how $N_{maj}$ guarantees tail-intersection collapse regardless of the relative variance $\sigma^2$ between the clusters.

---

### Official Review · Reviewer_4mxs · 2026-03-13

**Soundness:** 2
**Presentation:** 3
**Significance:** 2
**Originality:** 3
**Overall Recommendation:** 3
**Confidence:** 3

**Summary:**

This paper analyzes a problem in RAG systems where minority target documents become harder to retrieve as the number of majority documents increases. The paper studies this issue under two settings: a static setting where embeddings are fixed, and a dynamic setting where embeddings are continuously updated. In both cases, they analytically compute how many interfering documents are expected to fall within the distance between the query and the target document. Documents are modeled using a Poisson point process. In the static case, they show that the success decreases exponentially as the macroscopic population increases. When a shortlist size is imposed, the probability decays super-polynomially as the number of majority documents increases relative to the shortlist size. In the dynamic model, they assume an update rule defined by an energy term composed of interaction potential and entropy. Under this setting, they show that when the minority density is low, the document density converges to zero. Finally, they demonstrate this phenomenon empirically using both synthetic experiments and realistic scenarios.

**Compliance With Llm Reviewing Policy:**

Affirmed.

**Final Justification:**

The reviewer agrees that the paper indeed tackles an interesting problem worth tackling. It is also appealing that the paper attempts to explain the phenomenon from a mathematical perspective. However, as the reviewer initially pointed out, it is difficult to agree that these theoretical claims actually lead to surprising findings and due to the lack of a baseline for addressing the discovered issue, the reviewer believe there are limited takeaways. Therefore, the reviewer would like to maintain the original rating.

**Key Questions For Authors:**

- In the static analysis, if the distance between the minority target embedding and the query is sufficiently small, would the effect of majority interference become negligible even as the majority population increases?
- The phase transition where performance drops after a critical point is shown with synthetic data, but in the real data results (Figure 4) the performance seems to drop rapidly from the beginning. What explains this difference?
- The evaluation considers retrieval incorrect if the exact target is not found. However, in practice there may be majority embeddings that are very close to the target. For example, in CIFAR-100 (otter), an image with a similar background or pose might be retrieved instead of another otter image. In such cases, should this still be considered a failure of minority detection?

**Limitations:**

The paper did not discuss its limitations, so additional discussion on the limitations and future directions may help the community better understand the work.

**Strengths And Weaknesses:**

Strength
- The paper highlights a problem in shared embedding spaces that can frequently occur in topics such as RAG, hubness, and fairness, yet is easy to overlook.
- The phenomenon is defined and analyzed rigorously from a theoretical perspective, and the explanation is clearly written and easy to follow.
- The authors support their analysis with experiments on various datasets, showing that as the majority population increases, the minority retrieval success rate decreases in both vision and text settings.

Weakness
- The reviewer thinks there are several strong assumptions are made. For example, document locations are modeled as a Poisson point process, and the embedding space is treated as simple Euclidean geometry. The analysis also assumes only one closest minority target and ignores interactions with other nearby minority data. In the dynamic model, the learning process is also simplified without various regularization terms.
- Although the paper proposes a mathematical framework to analyze and understand the phenomenon, it is somewhat trivial that accuracy decreases when the number of non-target documents increases.
- While the paper clearly raises the problem, it does not propose a solution to solve the raised problem.
- Experiments on real data cover multiple domains, but the settings of individual experiments are limited. For example, only "otter" and "Noir" are investigated in the CIFAR100 and Movie plot data.

---

> ### Author Rebuttal · Authors · 2026-03-28
>
> We thank the reviewer for recognizing our theoretical rigor and empirical evaluation, and appreciate your insightful feedback. We address your specific questions below.
>
> **W2 "Triviality" of the Phenomenon** It might intuitively seem obvious that adding more documents increases the chance of retrieval error, but the nature and severity of this degradation are mathematically non-trivial. Our framework proves that performance does not degrade smoothly or proportionally; in fact, it undergoes a catastrophic, super-polynomial collapse once a critical density threshold is reached. Furthermore, our dynamic analysis shows an emergent mechanism where standard accuracy-maximizing objectives drive the system to self-organize and strictly exclude minority interests. This is not merely an expected accumulation of stochastic noise but a systemic and irreversible erasure of minority representational capacity.
>
> **W1 Assumptions of Point Processes, Geometry and Regularization** Regarding the uniform Poisson Point Process (PPP) assumption, Appendix B.1 shows it serves as a conservative lower bound compared to a realistic clustered Gaussian Mixture Model. Regarding Euclidean geometry, Appendix D addresses how highly anisotropic Transformer embeddings collapse into a narrow sub-manifold ("cone effect"). This actually exacerbates the goal collision problem; by reducing the space's usable capacity, these real-world constraints make collisions inevitable even in high dimensions. We will explicitly reference both of these practical geometries in Section 2.1. Finally, regarding the dynamic model, to clarify, Section 3.1 explicitly includes an entropic regularizer ($\mathcal{E}$, weighted by diffusion $D$) in the free energy functional. This term functions as a standard ML regularizer by capturing SGD noise and preventing point-mass collapse. In fact, Theorem 3.1 directly analyzes the competition between learning pressure ($K$) and this regularization ($D$). We will make this connection explicit in the text.
>
> **Q2 Real vs Synthetic Data and the Phase Transition** The more rapid performance drop in real data (Figure 4) compared to synthetic data is precisely explained by the clustering effect (Appendix B.1). Real-world documents form clusters, creating pockets of super-critical density that trigger collapse at much lower population sizes than uniform models predict (dropping the critical threshold from $N_{maj} \approx 200,000$ to $\approx 5000$). We will elevate this clustering discussion to the main text interpreting Figure 4.
>
> **Q1 Negligible Interference at Small Distances** Our framework shows that for any fixed, non-zero distance $\epsilon$, majority interference will not become negligible. In practice, natural linguistic variance and model noise strictly bound $\epsilon$ away from zero, so the retrieval ball volume $V_d(\epsilon)$ is always positive. As majority population $N_{maj}$ increases, local majority document intensity $\lambda_D(x) \approx N_{maj}\rho_{maj}(x)$ grows unbounded. Consequently, the expected interferers inside this finite ball inevitably exceed any fixed shortlist size $L$. As proven in Theorem 2.2, retrieval success probability will inevitably collapse to zero as $N_{maj} \to \infty$, regardless of how small the initial non-zero distance $\epsilon$ is.
>
> **Q3 Definition of Retrieval Failure** Retrieving a visually similar non-target (e.g., a different animal instead of an otter) is considered a failure in RAG, which requires reliable grounding on specific external evidence. If a user queries for a highly specific concept like an "otter" or a "Film Noir" narrative, returning a generic "Crime/Mystery" document due to spatial overlap erases the query's nuanced intent. This substitution exemplifies geometric crowding where dense majority data statistically excludes adjacent minority targets by monopolizing nearest-neighbor ranks.
>
> **W4 Scope of Individual Experiments** We deliberately isolated a single minority class per dataset to strictly control for internal cluster coherence and isolate geometric crowding. To demonstrate generalizability, we evaluated 3 additional CIFAR-100 classes (Bear, Turtle, Fox) and 3 additional Wikipedia Movie Plot genres (Western, Musical, Horror). Across all new classes, we observed the exact catastrophic performance collapse predicted by Theorem 2.2. These graphs will be added to the appendix.
>
> **W3 Solution for the Problem and Limitations**
> We agree that concrete mitigation strategies are the important next step. In this paper, our primary goal was to formally define and mathematically prove this hidden failure mode to establish a foundation for those solutions. As detailed in our response to Reviewer BAo6, we propose Metadata Pre-filtering. Filtering on structured tags (e.g., genre) prior to vector search constrains the query to an isolated subspace. We will add a "Limitations" section noting the absence of a tested mitigation and our single-target retrieval assumption.

---

> > ### Author Rebuttal · Reviewer_4mxs · 2026-04-06
> >
> > The reviewer appreciates the detailed rebuttal from the authors. The rebuttal addresses some of my concerns, and the reviewer agrees that several of the assumptions are reasonably justified.
> > However, there are still some questions not fully resolved. In particular, while the theoretical analysis is well-developed, the reviewer is not yet convinced that the observed collapse is as surprising as claimed. For example, in Figure 2, the performance appears to decrease over a relatively broad range rather than exhibiting a sharply defined phase transition. It would strengthen the paper if the authors could more clearly demonstrate what makes this behavior unexpected (e.g., by comparing with cases where such collapse does not occur or showing a more distinct difference in collapse dynamics).
> >
> > Additionally, the analysis still lacks discussion of interactions among multiple minority clusters, which seems important for understanding more realistic settings. This question may related to the review of BAo6 about one-to-one mapping, so the reviewer read the reply. The response suggests that relaxing the one-to-one assumption to multi-document relevance would further amplify the collapse. However, while the phenomenon may still occur under multiple ground-truth settings, it remains unclear whether the current analysis, which relies on a one-to-one assumption, can adequately explain such scenarios.

---

> > > ### Author Response · Authors · 2026-04-06
> > >
> > > We thank the reviewer for the follow-up and their insightful questions.
> > >
> > > 1. As the reviewer correctly observes, the performance in Figure 2 appears to decrease over a "relatively broad range" rather than exhibiting a discontinuous vertical drop. This is a standard finite-size effect. In statistical mechanics, perfect phase transitions (step functions) only occur in the thermodynamic limit ($N \to \infty, L \to \infty$). Because our simulated system has a finite shortlist ($L=20$) and finite target distance, finite-size scaling smooths the step function into a sigmoid. What makes this behavior unexpected and surprising is not the visual sharpness of the curve, but the mathematical functional form of the decay in success probability. In a standard retrieval system where interference is purely stochastic, one would expect the success probability to degrade linearly or polynomially proportional to $1/N_{maj}$. However, Theorem 2.2 proves that once the critical density $N_c$ is crossed, the success probability drops exponentially. The system goes from near-perfect retrieval to effectively zero retrieval capacity with only a marginal increase in majority density. We will include a baseline plotting expected polynomial degradation ($O(1/N)$) against our theoretical exponential collapse to make this distinction clear in the manuscript.
> > >
> > > 2. The reviewer rightly points out that our formal proof assumes a one-to-one mapping, and asks for formal justification that the collapse persists under multi-document relevance. We can formalize this extension directly within our existing Poisson Point Process (PPP) framework. Let a minority query have $T > 1$ valid target documents, and define success as retrieving at least one target in the top-$L$ shortlist. Let $\epsilon_1 \le \epsilon_2 \dots \le \epsilon_T$ be the distances from the query to these $T$ targets.
> > >
> > > To fail, the system must push all $T$ targets out of the shortlist. For the closest target (at distance $\epsilon_1$), the expected number of interferers is $\Lambda(x) = N_{maj} I(x)$. Following our derivation in Appendix A.1, the probability that the closest target is excluded is bounded by
> > >
> > > $$P(\text{Exclude Target 1}) \approx 1 - \exp(-N_{maj} I_{min} + \dots)$$
> > >
> > > Because the targets are ordered by distance, if the closest target is excluded by $L$ interferers, all targets further away ($\epsilon_i > \epsilon_1$) are strictly excluded by those exact same $L$ interferers. Therefore, the probability of complete failure (excluding all $T$ targets) reduces entirely to the probability of excluding the nearest target. Mathematically speaking, increasing the number of ground-truth targets $T$ only shrinks the distance to the 1st-order statistic $\epsilon_{1}$ which marginally reduces the retrieval ball volume $V_d(\epsilon_1)$. This slightly increases the critical threshold $N_c$ and delays the start of the collapse. But the dependence on $N_{maj}$ remains in the exponent. Once the new threshold $N_c$ is crossed, the probability of finding any of the $T$ targets still decays exponentially. The fundamental mechanism of collapse (geometric saturation of the local neighborhood) is mathematically invariant to $T$. We will add a corollary to Theorem 2.2 in the Appendix to provide details about this exact proof using order statistics, and show that multiple targets delay (but cannot prevent) emergent marginalization.

---

### Decision · Program_Chairs · 2026-04-30

**Decision:**

Accept (regular)

**Comment:**

This paper studies dense retrieval in retrieval-augmented agents, where minority content is missing in top-k results due to overcrowding from the majority interests.

### Strengths

- The paper addresses an important and timely problem, namely how shared embedding spaces can create system-level retrieval failures.
- The work is original in how it connects retrieval crowding, mean-field theory, and fairness/marginalization in a unified framework.
- Several reviewers found the core logic clear and the theoretical development rigorous, and the strongest reviewer viewed the contribution as genuinely novel and practically important.
- The paper supports the theory with a fairly broad empirical evaluation.

### Weaknesses

- The paper relies on several modeling assumptions, including the Poisson point process, one-to-one target relevance, and a stylized dynamic learning model.
- Some reviewers were not convinced that the main phenomenon is as surprising as the paper claims.
- The practical contribution is limited by the lack of concrete mitigation experiments.
- The discussion of closely related work, especially on popularity bias, bias amplification, and retrieval fairness, was initially limited.

### Rebuttal and Remaining Concerns

During the rebuttal, the authors clarified the role of the assumptions, argued that clustered distributions make the collapse even stronger, expanded the interpretation of the dynamic McKean-Vlasov analysis, addressed multi-document relevance, and committed to strengthening the related-work, limitations, and deployment discussion.

Two reviewers maintained weak-reject scores, mainly because they still viewed the assumptions as strong, the practical takeaways as limited without mitigation experiments. The paper deserves credit for addressing an important problem with a novel and technically serious framework, and I think several of the identified weaknesses are addressable through revision. However, in the current version, the combination of assumptions, limited mitigation evidence, and remaining questions about practical grounding make the clear acceptance recommendation not yet sufficiently convincing.